# Harmonizing Maximum Likelihood with GANs for Multimodal Conditional Generation

**Soochan Lee, Junsoo Ha & Gunhee Kim**
Department of Computer Science and Engineering, Seoul National University, Seoul, Korea
soochan.lee@vision.snu.ac.kr, junsooha@hanyang.ac.kr, gunhee@snu.ac.kr
http://vision.snu.ac.kr/projects/mr-gan

## Abstract

Recent advances in conditional image generation tasks, such as image-to-image translation and image inpainting, are largely accounted to the success of conditional GAN models, which are often optimized by the joint use of the GAN loss with the reconstruction loss. However, we reveal that this training recipe shared by almost all existing methods causes one critical side effect: lack of diversity in output samples. In order to accomplish both training stability and multimodal output generation, we propose novel training schemes with a new set of losses named *moment reconstruction losses* that simply replace the reconstruction loss. We show that our approach is applicable to any conditional generation tasks by performing thorough experiments on image-to-image translation, super-resolution and image inpainting using Cityscapes and CelebA dataset. Quantitative evaluations also confirm that our methods achieve a great diversity in outputs while retaining or even improving the visual fidelity of generated samples.

## 1 Introduction

Recently, active research has led to a huge progress on *conditional image generation*, whose typical tasks include image-to-image translation (Isola et al. (2017)), image inpainting (Pathak et al. (2016)), super-resolution (Ledig et al. (2017)) and video prediction (Mathieu et al. (2016)). At the core of such advances is the success of conditional GANs (Mirza & Osindero (2014)), which improve GANs by allowing the generator to take an additional code or condition to control the modes of the data being generated. However, training GANs, including conditional GANs, is highly unstable and easy to collapse (Goodfellow et al. (2014)). To mitigate such instability, almost all previous models in conditional image generation exploit the reconstruction loss such as $\ell_1/\ell_2$ loss in addition to the GAN loss. Indeed, using these two types of losses is synergetic in that the GAN loss complements the weakness of the reconstruction loss that output samples are blurry and lack high-frequency structure, while the reconstruction loss offers the training stability required for convergence.

In spite of its success, we argue that it causes one critical side effect; the reconstruction loss aggravates the mode collapse, one of notorious problems of GANs. In conditional generation tasks, which are to intrinsically learn one-to-many mappings, the model is expected to generate diverse outputs from a single conditional input, depending on some stochastic variables (*e.g.* many realistic street scene images for a single segmentation map (Isola et al., 2017)). Nevertheless, such stochastic input rarely generates any diversity in the output, and surprisingly many previous methods omit a random noise source in their models. Most papers rarely mention the necessity of random noise, and a few others report that the model completely ignores the noise even if it is fed into the model. For example, Isola et al. (2017) state that the generator simply learns to ignore the noise, and even dropout fails to incur meaningful output variation.

The objective of this paper is to propose a new set of losses named *moment reconstruction losses* that can replace the reconstruction loss with losing neither the visual fidelity nor diversity in output samples. The core idea is to use maximum likelihood estimation (MLE) loss (*e.g.* $\ell_1/\ell_2$ loss) to predict conditional statistics of the real data distribution instead of applying it directly to the generator as done in most existing algorithms. Then, we assist GAN training by enforcing the generated distribution to match its statistics to the statistics of the real distribution.

In summary, our major contributions are three-fold. First, we show that there is a significant mismatch between the GAN loss and the reconstruction loss, thereby the model cannot achieve the optimality w.r.t. both losses. Second, we propose two novel loss functions that enable the model to accomplish both training stability and multimodal output generation. Our methods simply replace the reconstruction loss, and thus are applicable to any conditional generation tasks. Finally, we show the effectiveness and generality of our methods through extensive experiments on three generation tasks, including image-to-image translation, super-resolution and image inpainting, where our methods outperform recent strong baselines in terms of realism and diversity.

## 2 RELATED WORKS

**Conditional Generation Tasks.** Since the advent of GANs (Goodfellow et al. (2014)) and conditional GANs (Mirza & Osindero (2014)), there has been a large body of work in conditional generation tasks. A non-exhaustive list includes *image translation* (Isola et al., 2017; Wang et al., 2017; 2018; Zhang et al., 2017b; Lyu et al., 2017; Sangkloy et al., 2017; Xian et al., 2017; Zhang et al., 2017a; Liu et al., 2017), *super-resolution* (Ledig et al., 2017; Xie et al., 2018; Mahapatra et al., 2017; Xu et al., 2017; Bulat et al., 2018; Sajjadi et al., 2017), *image inpainting* (Pathak et al., 2016; Iizuka et al., 2017; Ouyang et al., 2018; Olszewski et al., 2017; Yu et al., 2018; Sabini & Rusak, 2018; Song et al., 2018; Li et al., 2017; Yang et al., 2018; Yeh et al., 2017) and *video prediction* (Mathieu et al., 2016; Jang et al., 2018; Lu et al., 2017; Villegas et al., 2017; Zhou & Berg, 2016; Bhattacharjee & Das, 2017; Vondrick & Torralba, 2017).

However, existing models have one common limitation: lack of stochasticity for diverse output. Despite the fact that the tasks are to be one-to-many mapping, they ignore random noise input which is necessary to generate diverse samples from a single conditional input. A number of works such as (Mathieu et al., 2016; Isola et al., 2017; Xie et al., 2018) try injecting random noise into their models but discover that the models simply discard it and instead learn a deterministic mapping.

**Multimodality Enhancing Models.** It is not fully understood yet why conditional GAN models fail to learn the full multimodality of data distribution. Recently, there has been a series of attempts to incorporate stochasticity in conditional generation as follows.

(1) *Conditional VAE-GAN.* VAE-GAN (Larsen et al., 2016) is a hybrid model that combines the decoder in VAE (Kingma & Welling, 2014) with the generator in GAN (Goodfellow et al., 2014). Its conditional variants have been also proposed such as CVAE-GAN (Bao et al., 2017), Bicycle-GAN (Zhu et al., 2017) and SAVP (Lee et al., 2018a). These models harness the strengths of the two models, output fidelity by GANs and diversity by VAEs, to produce a wide range of realistic images. Intuitively, the VAE structure drives the generator to exploit latent variables to represent the multimodality of the conditional distribution.

(2) *Disentangled representation.* Huang et al. (2018) and Lee et al. (2018b) propose to learn disentangled representation for multimodal unsupervised image-to-image translation. These models split the embedding space into a domain-invariant space for sharing information across domains and a domain-specific space for capturing styles and attributes. These models encode an input into the domain-invariant embedding and sample domain-specific embedding from some prior distribution. The two embeddings are fed into the decoder of the target domain, for which the model can generate diverse samples.

Conditional VAE-GANs and disentanglement-based methods both leverage the latent variable to prevent the model from discarding the multimodality of output samples. On the other hand, we present a simpler and orthogonal direction to achieve multimodal conditional generation by introducing novel loss functions that can replace the reconstruction loss.

## 3 LOSS MISMATCH OF CONDITIONAL GANS

We briefly review the objective of conditional GANs in section 3.1 and discuss why the two loss terms cause the loss of modality in the sample distribution of the generator in section 3.2.

### 3.1 PRELIMINARY: THE OBJECTIVE OF CONDITIONAL GANS

Conditional GANs aims at learning to generate samples that are indistinguishable from real data for a given input. The objective of conditional GANs usually consists of two terms, the GAN loss $\mathcal{L}_{\text{GAN}}$ and the reconstruction loss $\mathcal{L}_{\text{Rec}}$.

$$\mathcal{L} = \mathcal{L}_{\text{GAN}} + \lambda_{\text{Rec}}\mathcal{L}_{\text{Rec}} \tag{1}$$

Another popular loss term is the perceptual loss (Johnson et al., 2016; Bruna et al., 2016; Ledig et al., 2017). While the reconstruction loss encodes the pixel-level distance, the perceptual loss is defined as the distance between the features encoded by neural networks. Since they share the same form (*e.g.* $\ell_1/\ell_2$ loss), we consider the perceptual loss as a branch of the reconstruction loss.

The loss $\mathcal{L}_{\text{GAN}}$ is defined to minimize some distance measure (*e.g.* JS-divergence) between the true and generated data distribution conditioned on input $x$. The training scheme is often formulated as the following minimax game between the discriminator $D$ and the generator $G$.

$$\min_G \max_D \mathbb{E}_{x,y}[\log D(x, y)] + \mathbb{E}_{x,z}[\log(1 - D(x, G(x, z)))] \tag{2}$$

where each data point is a pair $(x, y)$, and $G$ generates outputs given an input $x$ and a random noise $z$. Note that $D$ also observes $x$, which is crucial for the performance (Isola et al., 2017).

The most common reconstruction losses in conditional GAN literature are the $\ell_1$ (Isola et al., 2017; Wang et al., 2017) and $\ell_2$ loss (Pathak et al., 2016; Mathieu et al., 2016). Both losses can be formulated as follows with $p = 1, 2$, respectively.

$$\mathcal{L}_{\text{Rec}} = \mathcal{L}_p = \mathbb{E}_{x,y,z}[\|y - G(x, z)\|_p^p]. \tag{3}$$

These two losses naturally stem from the maximum likelihood estimations (MLEs) of the parameters of Laplace and Gaussian distribution, respectively. The likelihood of dataset $(\mathbb{X}, \mathbb{Y})$ assuming each distribution is defined as follows.

$$p_{\text{L}}(\mathbb{Y}|\mathbb{X}; \boldsymbol{\theta}) = \prod_{i=1}^{N} \frac{1}{2b}\exp(-\frac{|y_i - f_{\boldsymbol{\theta}}(x_i)|}{b}), \; p_{\text{G}}(\mathbb{Y}|\mathbb{X}; \boldsymbol{\theta}) = \prod_{i=1}^{N} \frac{1}{\sqrt{2\pi\sigma^2}}\exp(-\frac{(y_i - f_{\boldsymbol{\theta}}(x_i))^2}{2\sigma^2})$$

where $N$ is the size of the dataset, and $f$ is the model parameterized by $\boldsymbol{\theta}$. The central and dispersion measure for Gaussian are the mean and variance $\sigma^2$, and the correspondences for Laplace are the median and mean absolute deviation (MAD) $b$. Therefore, using $\ell_2$ loss leads the model output $f_{\boldsymbol{\theta}}(x)$ to become an estimate of the conditional average of $y$ given $x$, while using $\ell_1$ loss is equivalent to estimating the conditional median of $y$ given $x$ (Bishop, 2006). Note that the model $f_{\boldsymbol{\theta}}$ is trained to predict the mean (or median) of the data distribution, not to generate samples from the distribution.

### 3.2 LOSS OF MODALITY BY THE RECONSTRUCTION LOSS

We argue that the joint use of the reconstruction loss with the GAN loss can be problematic, because it may worsen the mode collapse. Below, we discuss this argument both mathematically and empirically.

One problem of the $\ell_2$ loss is that it forces the model to predict only the mean of $p(y|x)$, while pushing the conditional variance to zero. According to James (2003), for any symmetric loss function $\mathcal{L}_s$ and an estimator $\hat{y}$ for $y$, the loss is decomposed into one irreducible term $\text{Var}(y)$ and two reducible terms, $\text{SE}(\hat{y}, y)$ and $\text{VE}(\hat{y}, y)$, where SE refers to *systematic effect*, the change in error caused by the bias of the output, while VE refers to *variance effect*, the change in error caused by the variance of the output.

$$\mathbb{E}_{y,\hat{y}}[\mathcal{L}_s(y, \hat{y})] = \underbrace{\mathbb{E}_y\big[\mathcal{L}_s(y, Sy)\big]}_{\text{Var}(y)} + \underbrace{\mathbb{E}_y\big[\mathcal{L}_s(y, S\hat{y}) - \mathcal{L}_s(y, Sy)\big]}_{\text{SE}(\hat{y},y)}$$
$$+ \underbrace{\mathbb{E}_{y,\hat{y}}\big[\mathcal{L}_s(y, \hat{y}) - \mathcal{L}_s(y, S\hat{y})\big]}_{\text{VE}(\hat{y},y)}, \tag{4}$$

where $S$ is an operator that is defined to be

$$Sy = \arg\min_{\mu} \mathbb{E}_y[\mathcal{L}_s(y, \mu)].$$

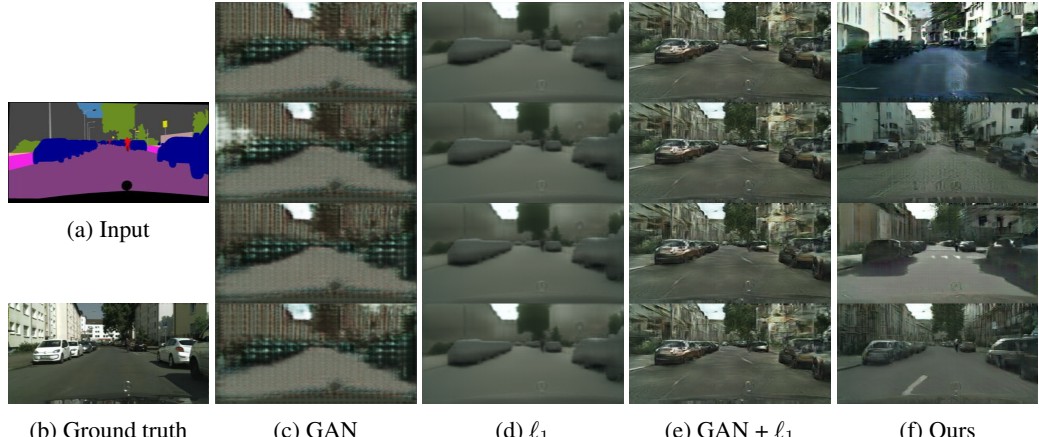

| (a) Input | | | | |
| (b) Ground truth | (c) GAN | (d) $\ell_1$ | (e) GAN + $\ell_1$ | (f) Ours |

Figure 1: The loss mismatch in conditional GANs with examples of a Pix2Pix variant trained on Cityscapes dataset. We compare the results according to different loss functions (columns) with different noise input (rows). (c) Using the GAN loss alone is not enough to learn the complex data distribution due to its training instability. (d) Using the $\ell_1$ loss only, the model generates images fairly close to the ground-truth, but they are far from being realistic. (e) The combination of the two losses enables the model to generate visually appealing samples. However, the model completely ignores noise input and generates almost identical images. (f) In contrast, the model trained with our loss term (*i.e.* proxy $MR_2$) generates a diverse set of images fully utilizing noise input.

Notice that the total loss is minimized when $\hat{y} = S\hat{y} = Sy$, reducing both SE and VE to 0. For $\ell_2$ loss, $Sy$ and $S\hat{y}$ are the expectations of $y$ and $\hat{y}$. Furthermore, SE and VE correspond to the squared bias and the variance, respectively.

$$\text{SE}(\hat{y}, y) = \mathbb{E}_y\big[(y - S\hat{y})^2 - (y - Sy)^2\big] = (S\hat{y} - Sy)^2, \tag{5}$$

$$\text{VE}(\hat{y}, y) = \mathbb{E}_{y,\hat{y}}\big[(y - \hat{y})^2 - (y - S\hat{y})^2\big] = \mathbb{E}_{\hat{y}}\big[(\hat{y} - S\hat{y})^2\big]. \tag{6}$$

The decomposition above demonstrates that the minimization of $\ell_2$ loss is equivalent to the minimization of the bias and the variance of prediction. In the context of conditional generation tasks, this implies that $\ell_2$ loss minimizes the conditional variance of output.

Figure 1 shows some examples of the Pix2Pix model (Isola et al. (2017)) that is applied to translate segmentation labels to realistic photos in Cityscapes dataset (Cordts et al., 2016). We slightly modify the model so that it takes additional noise input. We use $\ell_1$ loss as the reconstruction loss as done in the original paper. We train four models that use different combinations of loss terms, and generate four samples with different noise input. As shown in Figure 1(c), the model trained with only the GAN loss fails to generate realistic images, since the signal from the discriminator is too unstable to learn the translation task. In Figure 1(d), the model with only $\ell_1$ loss is more stable but produces results far from being realistic. The combination of the two losses in Figure 1(e) helps not only to reliably train the model but also to generate visually appealing images; yet, its results lack variation. This phenomenon is also reported in (Mathieu et al., 2016; Isola et al., 2017), although the cause is unknown. Pathak et al. (2016) and Iizuka et al. (2017) even state that better results are obtained without noise in the models. Finally, Figure 1(f) shows that our new objective enables the model to generate not only visually appealing but also diverse output samples.

## 4 APPROACH

We propose novel alternatives for the reconstruction loss that are applicable to virtually any conditional generation tasks. Trained with our new loss terms, the model can accomplish both the stability of training and multimodal generation as already seen in Figure 1(f). Figure 2 illustrates architectural comparison between conventional conditional GANs and our models with the new loss terms. In conventional conditional GANs, MLE losses are applied to the generator's objective to make sure that it generates output samples well matched to their ground-truth. On the other hand,

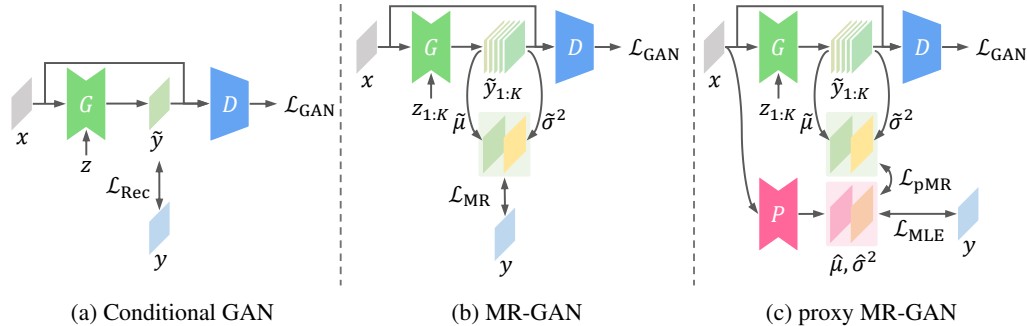

(a) Conditional GAN      (b) MR-GAN      (c) proxy MR-GAN

Figure 2: The architecture comparison of the proposed MR-GAN and proxy MR-GAN with conventional conditional GANs. (a) In conditional GANs, $G$ generates a sample $\tilde{y}$ (*e.g.* a realistic image) from an input $x$ (*e.g.* a segmentation map) and $D$ determines whether $\tilde{y}$ is real or fake. The reconstruction loss $\mathcal{L}_{\mathrm{Rec}}$ enforces a sample $\tilde{y}$ to be similar to a ground-truth image $y$. (b) In our first model MR-GAN, $G$ generates a set of samples $\tilde{y}_{1:K}$ to calculate the sample moments, *i.e.* mean and variance. Then, we compute the moment reconstruction loss, which is basically an MLE loss between the sample moments and $y$. (c) The second model proxy MR-GAN is a more stable variant of MR-GAN. The predictor $P$, a twin network of $G$, is trained with an MLE loss to estimate the conditional mean and variances of $y$ given $x$. Then, we match the sample mean $\tilde{\mu}$ to the predicted $\hat{\mu}$ and the sample variance $\tilde{\sigma}^2$ to the predicted $\hat{\sigma}^2$.

our key idea is to apply MLE losses to make sure that the *statistics*, such as mean or variance, of the conditional distribution $p(y|x)$ are similar between the generator's sample distribution and the actual data distribution.

In section 4.1, we extend the $\ell_1$ and $\ell_2$ loss to estimate all the parameters of Laplace and Gaussian distribution, respectively. In section 4.2–4.3, we propose two novel losses for conditional GANs. Finally, in section 4.4, we show that the reconstruction loss conflicts with the GAN loss while our losses do not.

## 4.1 THE MLE FOR MEAN AND VARIANCE

The $\ell_2$ loss encourages the model to perform the MLE of the conditional mean of $y$ given $x$ while the variance $\sigma^2$, the other parameter of Gaussian, is assumed to be fixed. If we allow the model to estimate the conditional variance as well, the MLE loss corresponds to

$$\mathcal{L}_{\mathrm{MLE,Gaussian}} = \mathbb{E}_{x,y}\left[\frac{(y-\hat{\mu})^2}{2\hat{\sigma}^2} + \frac{1}{2}\log\hat{\sigma}^2\right] \quad \text{where } \hat{\mu}, \hat{\sigma}^2 = f_{\boldsymbol{\theta}}(x) \tag{7}$$

where the model $f_{\boldsymbol{\theta}}$ now estimates both $\hat{\mu}$ and $\hat{\sigma}^2$ for $x$. Estimating the conditional variance along with the mean can be interpreted as estimating the heteroscedastic aleatoric uncertainty in Kendall & Gal (2017) where the variance is the measure of aleatoric uncertainty.

For the Laplace distribution, we can derive the similar MLE loss as

$$\mathcal{L}_{\mathrm{MLE,Laplace}} = \mathbb{E}_{x,y}\left[\frac{|y-\hat{m}|}{\hat{b}} + \log\hat{b}\right] \quad \text{where } \hat{m}, \hat{b} = f_{\boldsymbol{\theta}}(x) \tag{8}$$

where $\hat{m}$ is the predicted median and $\hat{b}$ is the predicted MAD (Bloesch et al., 2018). In practice, it is more numerically stable to predict the logarithm of variance or MAD (Kendall & Gal, 2017).

In the following, we will describe our methods mainly with the $\ell_2$ loss under Gaussian assumption. It is straightforward to obtain the Laplace version of our methods for the $\ell_1$ loss by simply replacing the mean and variance with the median and MAD.

## 4.2 THE MOMENT RECONSTRUCTION LOSS

Our first loss function is named *Moment Reconstruction (MR)* loss, and we call a conditional GAN with the loss as *MR-GAN*. As depicted in Figure 2(b), the overall architecture of MR-GAN follows

that of a conditional GAN, but there are two important updates. First, the generator produces $K$ different samples $\tilde{y}_{1:K}$ for each input $x$ by varying noise input $z$. Second, the MLE loss is applied to the sample moments (*i.e.* mean and variance) in contrast to the reconstruction loss which is used directly on the samples. Since the generator is supposed to approximate the real distribution, we can regard that the moments of the generated distribution estimate the moments of the real distribution. We approximate the moments of the generated distribution with the sample moments $\tilde{\mu}$ and $\tilde{\sigma}^2$, which are defined as

$$\tilde{\mu} = \frac{1}{K}\sum_{i=1}^{K}\tilde{y}_i, \ \ \tilde{\sigma}^2 = \frac{1}{K-1}\sum_{i=1}^{K}(\tilde{y}_i - \tilde{\mu})^2, \ \ \text{where} \ \ \tilde{y}_{1:K} = G(x, z_{1:K}). \tag{9}$$

The MR loss is defined by plugging $\tilde{\mu}$ and $\tilde{\sigma}^2$ in Eq.(9) into $\hat{\mu}$ and $\hat{\sigma}^2$ in Eq.(7). The final loss of MR-GAN is the weighted sum of the GAN loss and the MR loss:

$$\mathcal{L} = \mathcal{L}_{\text{GAN}} + \lambda_{\text{MR}}\mathcal{L}_{\text{MR}}. \tag{10}$$

As a simpler variant, we can use only $\hat{\mu}$ and ignore $\hat{\sigma}^2$. We denote this simpler one as $\text{MR}_1$ (*i.e.* using mean only) and the original one as $\text{MR}_2$ (*i.e.* using both mean and variance) according to the number of moments used.

$$\mathcal{L}_{\text{MR}_1} = \mathbb{E}_{x,y}\big[|y - \tilde{\mu}|^2\big], \mathcal{L}_{\text{MR}_2} = \mathbb{E}_{x,y}\left[\frac{(y - \tilde{\mu})^2}{2\tilde{\sigma}^2} + \frac{1}{2}\log\tilde{\sigma}^2\right]. \tag{11}$$

In addition, we can easily derive the Laplace versions of the MR loss that use median and MAD.

### 4.3 THE PROXY MOMENT RECONSTRUCTION LOSS

One possible drawback of the MR loss is that the training can be unstable at the early phase, since the irreducible noise in $y$ directly contributes to the total loss. For more stable training, we propose a variant called *Proxy Moment Reconstruction (proxy MR) loss* and a conditional GAN with the loss named *proxy MR-GAN* whose overall training scheme is depicted in Figure 2(c). The key difference is the presence of *predictor P*, which is a clone of the generator with some minor differences: (i) no noise source as input and (ii) prediction of both mean and variance as output. The predictor is trained prior to the generator by the MLE loss in Eq.(7) with ground-truth $y$ to predict conditional mean and variance, *i.e.* $\hat{\mu}$ and $\hat{\sigma}^2$. When training the generator, we utilize the predicted statistics of real distribution to guide the outputs of the generator. Specifically, we match the predicted mean/variance and the mean/variance of the generator's distribution, which is computed by the sample mean $\tilde{\mu}$ and variance $\tilde{\sigma}^2$ from $\tilde{y}_{1:K}$. Then, we define the proxy MR loss $\mathcal{L}_{\text{pMR}}$ as the sum of squared errors between predicted statistics and sample statistics:

$$\mathcal{L}_{\text{pMR}} = (\tilde{\mu} - \hat{\mu})^2 + (\tilde{\sigma}^2 - \hat{\sigma}^2)^2 \ \text{where} \ \hat{\mu}, \hat{\sigma}^2 = P(x). \tag{12}$$

One possible variant is to match only the first moment $\mu$. As with the MR loss, we denote the original method proxy $\text{MR}_2$ and the variant proxy $\text{MR}_1$ where the number indicates the number of matched moments. Deriving the Laplace version of the proxy MR loss is also straightforward. The detailed algorithms of all eight variants are presented in appendix A.

Compared to proxy MR-GAN, MR-GAN allows the generator to access real data $y$ directly; thus there is no bias caused by the predictor. On the other hand, the use of predictor in proxy MR-GAN provides less variance in target values and leads more stable training especially when the batch or sample size is small. Another important aspect worth comparison is overfitting, which should be carefully considered when using MLE with finite training data. In proxy MR-GAN, we can choose the predictor with the smallest validation loss to avoid overfitting, and freeze it while training the generator. Therefore, the generator trained with the proxy MR loss suffers less from overfitting, compared to that trained with the MR loss that directly observes training data. To sum up, the two methods have their own pros and cons. We will empirically compare the behaviors of these two approaches in section 5.2.

### 4.4 ANALYSES

We here show the mismatch between the GAN loss and the $\ell_2$ loss in terms of the set of optimal generators. We then discuss why our approach does not suffer from the loss of diversity in the output samples. Refer to appendix E for the mismatch of $\ell_1$ loss.

The sets of optimal generators for the GAN loss and the $\ell_2$ loss, denoted by $\mathbb{G}$ and $\mathbb{R}$ respectively, can be formulated as follows:

$$\mathbb{G} = \{G | p_{\text{data}}(y|x) = p_G(y|x)\}, \quad \mathbb{R} = \{G | \arg\min_G \mathbb{E}_{x,y,z}[\mathcal{L}_2(G(x,z), y)]\}. \quad (13)$$

Recall that $\ell_2$ loss is minimized when both bias and variance are zero, which implies that $\mathbb{R}$ is a subset of $\mathbb{V}$ where $G$ has no conditional variance:

$$\mathbb{R} \subset \mathbb{V} = \{G | \text{Var}_z(G(x,z)|x) = 0\}. \quad (14)$$

In conditional generation tasks, however, the conditional variance $\text{Var}(y|x)$ is assumed to be non-zero (*i.e.* diverse output $y$ for a given input $x$). Thereby we conclude $\mathbb{G} \cap \mathbb{V} = \emptyset$, which reduces to $\mathbb{G} \cap \mathbb{R} = \emptyset$. It means that any generator $G$ cannot be optimal for both GAN and $\ell_2$ loss simultaneously. Therefore, it is hard to anticipate what solution is attained by training and how the model behaves when the two losses are combined. One thing for sure is that the reconstruction loss alone is designed to provide a sufficient condition for mode collapse as discussed in section 3.2.

Now we discuss why our approach does not suffer from loss of diversity in the output samples unlike the reconstruction loss in terms of the set of optimal generators. The sets of optimal mappings for our loss terms are formulated as follows:

$$\mathbb{M}_1 = \{G | \mathbb{E}_x[(\mathbb{E}_y[y|x] - \mathbb{E}_z[G(x,z)|x])^2] = 0\} \quad (15)$$

$$\mathbb{M}_2 = \{G | \mathbb{E}_x[(\mathbb{E}_y[y|x] - \mathbb{E}_z[G(x,z)|x])^2 + (\text{Var}_y(y|x) - \text{Var}_z(G(x,z)|x))^2] = 0\} \quad (16)$$

where $\mathbb{M}_1$ corresponds to MR$_1$ and proxy MR$_1$ while $\mathbb{M}_2$ corresponds to MR$_2$ and proxy MR$_2$ . It is straightforward to show that $\mathbb{G} \subset \mathbb{M}_2 \subset \mathbb{M}_1$; if $p_{\text{data}}(y|x) = p_G(y|x)$ is satisfied at optimum, the conditional expectations and variations of both sides should be the same too:

$$p_{\text{data}}(y|x) = p_G(y|x) \implies \big[\mathbb{E}_y[y|x] = \mathbb{E}_z[G(x,z)|x]\big] \wedge \big[\text{Var}_y(y|x) = \text{Var}_z(G(x,z)|x)\big]. \quad (17)$$

To summarize, there is no generator that is both optimal for the GAN loss and the $\ell_2$ loss since $\mathbb{G} \cap \mathbb{R} = \emptyset$. Moreover, as the $\ell_2$ loss pushes the conditional variance to zero, the final solution is likely to lose multimodality. On the other hand, the optimal generator w.r.t. the GAN loss is also optimal w.r.t. our loss terms since $\mathbb{G} \subset \mathbb{M}_2 \subset \mathbb{M}_1$.

This proof may not fully demonstrate why our approach does not give up multimodality, which could be an interesting future work in line with the mode collapsing issue of GANs. Nonetheless, we can at least assert that our loss functions do not suffer from the same side effect as the reconstruction loss does. Another remark is that this proof is confined to conditional GAN models that have no use of latent variables. It may not be applicable to the models that explicitly encode latent variables such as Bao et al. (2017), Zhu et al. (2017), Lee et al. (2018a), Huang et al. (2018), and Lee et al. (2018b), since the latent variables can model meaningful information about the target diversity.

## 5 EXPERIMENTS

In order to show the generality of our methods, we apply them to three conditional generation tasks: image-to-image translation, super-resolution and image inpainting, for each of which we select Pix2Pix, SRGAN (Ledig et al., 2017) and GLCIC (Iizuka et al., 2017) as base models, respectively. We use Maps (Isola et al., 2017) and Cityscapes dataset (Cordts et al., 2016) for image translation and CelebA dataset (Liu et al., 2015) for the other tasks. We minimally modify the base models to include random noise input and train them with MR and proxy MR objectives. We do not use any other loss terms such as perceptual loss, and train the models from scratch. We present the details of training and implementation and more thorough results in the appendix.

### 5.1 QUALITATIVE EVALUATION

In every task, our methods successfully generate diverse images as presented in Figure 3. From qualitative aspects, one of the most noticeable differences between MR loss and proxy MR loss lies in the training stability. We find that proxy MR loss works as stably as the reconstruction loss in all three tasks. On the other hand, we cannot find any working configuration of MR loss in the image

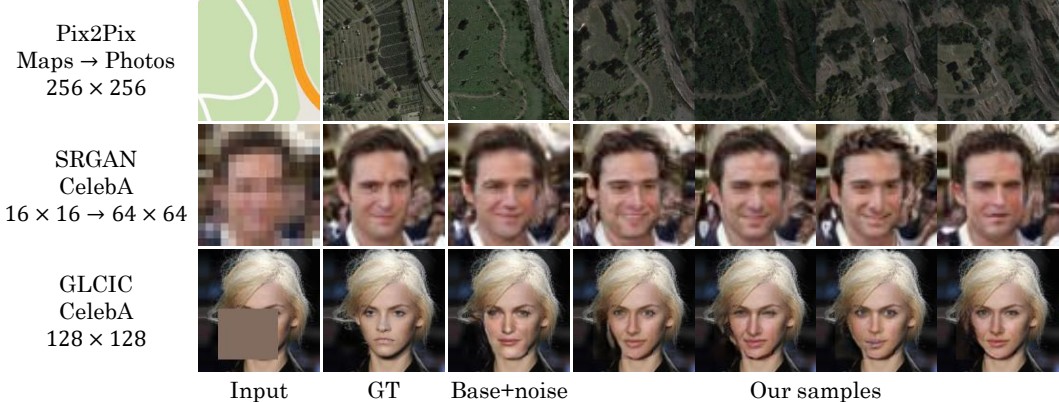

Figure 3: Comparison between the results of our proxy MR$_2$-GAN and the state-of-the-art methods on image-to-image translation, super-resolution and image inpainting tasks. In every task, our model generates diverse images of high quality, while existing methods with the reconstruction loss do not.

inpainting task. Also, the MR loss is more sensitive to the number of samples $K$ generated for each input. In SRGAN experiments, for example, both methods converge reliably with $K = 24$, while the MR loss often diverges at $K = 12$. Although the MR loss is simpler and can be trained in an end-to-end manner, its applicability is rather limited compared to the proxy MR loss. We provide generated samples for each configuration in the appendix.

## 5.2 QUANTITATIVE EVALUATION

Following Zhu et al. (2017), we quantitatively measure diversity and realism of generated images. We evaluate our methods on Pix2Pix–Cityscapes, SRGAN–CelebA and GLCIC–CelebA tasks. For each (method, task) pair, we generate 20 images from each of 300 different inputs using the trained model. As a result, the test sample set size is 6,000 in total. For diversity, we measure the average LPIPS score (Zhang et al., 2018). Among 20 generated images per input, we randomly choose 10 pairs of images and compute conditional LPIPS values. We then average the scores over the test set. For realism, we conduct a human evaluation experiment from 33 participants. We present a real or fake image one at a time and ask participants to tag whether it is real or fake. The images are presented for 1 second for SRGAN/GLCIC and 0.5 second for Pix2Pix, as done in Zhu et al. (2017). We calculate the accuracy of identifying fake images with averaged F-measure $F$ and use $2(1 - F)$ as the realism score. The score is assigned to 1.0 when all samples are completely indistinguishable from real images and 0 when the evaluators make no misclassification.

There are eight configurations of our methods in total, depending on the MLE (Gaussian and Laplace), the number of statistics (one and two) and the loss type (MR and proxy MR). We test all variants for the Pix2Pix task and only Gaussian proxy MR$_2$ loss for the others. We compare with base models in every task and additionally BicycleGAN (Zhu et al., 2017) for the Pix2Pix task. We use the official implementation by the authors with minimal modification. BicycleGAN has not been applied to image inpainting and super-resolution tasks, for which we do not report its result.

Table 1 summarizes the results. In terms of both realism and diversity, our methods achieve competitive performance compared to the base models, sometimes even better. For instance, in Pix2Pix–Cityscapes, three of our methods, Gaussian MR$_1$, Gaussian proxy MR$_1$, and Gaussian proxy MR$_2$, significantly outperform BicycleGAN and Pix2Pix+noise in both measures. Without exception, the diversity scores of our methods are far greater than those of the baselines while maintaining competitive realism scores. These results confirm that our methods generate a broad spectrum of high-quality images from a single input. Interestingly, MR$_1$ and proxy MR$_1$ loss generate comparable or even better outputs compared to MR$_2$ and proxy MR$_2$ loss. That is, matching means could be enough to guide GAN training in many tasks. It implies that adding more statistics (*i.e.* adding more *guidance* signal to the model) may be helpful in some cases, but generally may not improve the performance. Moreover, additional errors may arise from predicting more statistics, which could degrade the GAN training.

| Method | | Realism | Diversity |
|---|---|---|---|
| Random real data | | 1.00 | 0.559 |
| Pix2Pix+noise | | $0.22^{\pm 0.04}$ | 0.004 |
| BicycleGAN | | $0.16^{\pm 0.03}$ | 0.191 |
| Gaussian | $MR_1$ | $\mathbf{0.54}^{\pm 0.05}$ | 0.299 |
| | $MR_2$ | $0.14^{\pm 0.02}$ | 0.453 |
| | proxy $MR_1$ | $0.49^{\pm 0.05}$ | **0.519** |
| | proxy $MR_2$ | $0.44^{\pm 0.05}$ | 0.388 |
| Laplace | $MR_1$ | $0.19^{\pm 0.03}$ | 0.393 |
| | $MR_2$ | $0.20^{\pm 0.04}$ | 0.384 |
| | proxy $MR_1$ | $0.19^{\pm 0.03}$ | 0.368 |
| | proxy $MR_2$ | $0.17^{\pm 0.02}$ | 0.380 |

(a) Pix2Pix–Cityscapes

| Method | Realism | Diversity |
|---|---|---|
| Random real data | 1.00 | 0.290 |
| SRGAN+noise | $\mathbf{0.60}^{\pm 0.06}$ | 0.005 |
| Gaussian proxy $MR_2$ | $0.52^{\pm 0.05}$ | **0.050** |

(b) SRGAN–CelebA

| Method | Realism | Diversity |
|---|---|---|
| Random real data | 1.00 | 0.426 |
| GLCIC+noise | $0.28^{\pm 0.04}$ | 0.004 |
| Gaussian proxy $MR_2$ | $\mathbf{0.33}^{\pm 0.04}$ | **0.060** |

(c) GLCIC–CelebA

Table 1: Quantitative evaluation on three (method, dataset) pairs. Realism is measured by $2(1 - F)$ where $F$ is the averaged F-measure of identifying fake by human evaluators. Diversity is scored by the average of conditional LPIPS values. In both metrics, the higher the better. In all three tasks, our methods generate highly diverse images with competitive or even better realism.

## 6 CONCLUSION

In this work, we pointed out that there is a mismatch between the GAN loss and the conventional reconstruction losses. As alternatives, we proposed a set of novel loss functions named MR loss and proxy MR loss that enable conditional GAN models to accomplish both stability of training and multimodal generation. Empirically, we showed that our loss functions were successfully integrated with multiple state-of-the-art models for image translation, super-resolution and image inpainting tasks, for which our method generated realistic image samples of high visual fidelity and variability on Cityscapes and CelebA dataset.

There are numerous possible directions beyond this work. First, there are other conditional generation tasks that we did not cover, such as text-to-image synthesis, text-to-speech synthesis and video prediction, for which our methods can be directly applied to generate diverse, high-quality samples. Second, in terms of statistics matching, our methods can be extended to explore other higher order statistics or covariance. Third, using the statistics of high-level features may capture additional correlations that cannot be represented with pixel-level statistics.

### ACKNOWLEDGMENTS

We especially thank Insu Jeon for his valuable insight and helpful discussion. This work was supported by Video Analytics CoE of Security Labs in SK Telecom and Basic Science Research Program through National Research Foundation of Korea (NRF) (2017R1E1A1A01077431). Gunhee Kim is the corresponding author.

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

## A  ALGORITHMS

We elaborate on the algorithms of all eight variants of our methods in detail from Algorithm 5 to Algorithm 4. The presented algorithms assume a single input per update, although we use mini-batch training in practice. Also, we use non-saturating GAN loss, $-\log D(x, \tilde{y})$ (Goodfellow, 2016).

For Laplace MLEs, the statistics that we compute are median and MAD. Unlike mean, however, the gradient of median is defined only in terms of the single median sample. Therefore, a naive implementation would only calculate the gradient for the median sample, which is not effective for training. Therefore, we use a special trick to distribute the gradients to every sample. In proxy MR loss, we first calculate the difference between the predicted median and the sample median, and then add it to samples $\tilde{y}_{1:K}$ to set the target values $t_{1:K}$. We consider $t_{1:K}$ as constants so that the gradient is not calculated for the target values (this is equivalent to `Tensor.detach()` in PyTorch and `tf.stop_gradient()` in TensorFlow). Finally, we calculate the loss between the target values and samples, not the medians. We use the similar trick for the MR loss.

---

**Algorithm 1** Generator update in Gaussian $MR_1$-GAN

---

**Require:** Generator $G$, discriminator $D$, MR loss coefficient $\lambda_{\mathrm{MR}}$
**Require:** input $x$, ground truth $y$, the number of samples $K$
1: **for** $i = 1$ **to** $K$ **do**
2:     $z_i \leftarrow$ GenerateNoise()
3:     $\tilde{y}_i \leftarrow G(x, z_i)$ {Generate $K$ samples}
4: **end for**
5: $\tilde{\mu} \leftarrow \frac{1}{K} \sum_{i=1}^{K} \tilde{y}_i$ {Sample mean}
6: $\mathcal{L}_{\mathrm{MR}} \leftarrow (y - \tilde{\mu})^2$
7: $\mathcal{L}_{\mathrm{GAN}} \leftarrow \frac{1}{K} \sum_{i=1}^{K} -\log D(x, y_i)$
8: $\theta_G \leftarrow$ Optimize($\theta_G, \nabla_{\theta_G} \mathcal{L}_{\mathrm{GAN}} + \lambda_{\mathrm{MR}} \mathcal{L}_{\mathrm{MR}}$)

---

**Algorithm 2** Generator update in Gaussian $MR_2$-GAN

---

**Require:** Generator $G$, discriminator $D$, MR loss coefficient $\lambda_{\mathrm{MR}}$
**Require:** input $x$, ground truth $y$, the number of samples $K$
1: **for** $i = 1$ **to** $K$ **do**
2:     $z_i \leftarrow$ GenerateNoise()
3:     $\tilde{y}_i \leftarrow G(x, z_i)$ {Generate $K$ samples}
4: **end for**
5: $\tilde{\mu} \leftarrow \frac{1}{K} \sum_{i=1}^{K} \tilde{y}_i$ {Sample mean}
6: $\tilde{\sigma}^2 \leftarrow \frac{1}{K-1} \sum_{i=1}^{K} (\tilde{y}_i - \tilde{\mu})^2$ {Sample variance}
7: $\mathcal{L}_{\mathrm{MR}} \leftarrow \frac{(y-\tilde{\mu})^2}{2\tilde{\sigma}^2} + \frac{1}{2} \log \tilde{\sigma}^2$
8: $\mathcal{L}_{\mathrm{GAN}} \leftarrow \frac{1}{K} \sum_{i=1}^{K} -\log D(x, y_i)$
9: $\theta_G \leftarrow$ Optimize($\theta_G, \nabla_{\theta_G} \mathcal{L}_{\mathrm{GAN}} + \lambda_{\mathrm{MR}} \mathcal{L}_{\mathrm{MR}}$)

---

**Algorithm 3** Generator update in Laplace $MR_1$-GAN

---

**Require:** Generator $G$, discriminator $D$, MR loss coefficient $\lambda_{\mathrm{MR}}$
**Require:** input $x$, ground truth $y$, the number of samples $K$
1: **for** $i = 1$ **to** $K$ **do**
2:     $z_i \leftarrow$ GenerateNoise()
3:     $\tilde{y}_i \leftarrow G(x, z_i)$ {Generate $K$ samples}
4: **end for**
5: $\tilde{m} \leftarrow \mathrm{med}(\tilde{y}_{1:K})$ {Sample median}
6: $t_{1:K} = \mathrm{detach}(\tilde{y}_{1:K} + (y - \tilde{m}))$
7: $\mathcal{L}_{\mathrm{MR}} \leftarrow \frac{1}{K} \sum_{i=1}^{K} |t_i - \tilde{y}_i|$
8: $\mathcal{L}_{\mathrm{GAN}} \leftarrow \frac{1}{K} \sum_{i=1}^{K} -\log D(x, y_i)$
9: $\theta_G \leftarrow$ Optimize($\theta_G, \nabla_{\theta_G} \mathcal{L}_{\mathrm{GAN}} + \lambda_{\mathrm{MR}} \mathcal{L}_{\mathrm{MR}}$)

---

---

**Algorithm 4** Generator update in Laplace MR$_2$-GAN

---

**Require:** Generator $G$, discriminator $D$, MR loss coefficient $\lambda_{\mathrm{MR}}$
**Require:** input $x$, ground truth $y$, the number of samples $K$
 1: **for** $i = 1$ **to** $K$ **do**
 2:    $z_i \leftarrow \mathrm{GenerateNoise}()$
 3:    $\tilde{y}_i \leftarrow G(x, z_i)$ {Generate $K$ samples}
 4: **end for**
 5: $\tilde{m} \leftarrow \mathrm{med}(\tilde{y}_{1:K})$ {Sample median}
 6: $\tilde{b} \leftarrow \frac{1}{K} \sum_{i=1}^{K} |\tilde{y}_i - \tilde{m}|$ {Sample MAD}
 7: $t_{1:K} = \mathrm{detach}(\tilde{y}_{1:K} + (y - \tilde{m}))$
 8: $\mathcal{L}_{\mathrm{MR}} \leftarrow \frac{1}{K} \sum_{i=1}^{K} \frac{|t_i - \tilde{y}_i|}{\tilde{b}} + \log \tilde{b}$
 9: $\mathcal{L}_{\mathrm{GAN}} \leftarrow \frac{1}{K} \sum_{i=1}^{K} -\log D(x, y_i)$
10: $\theta_G \leftarrow \mathrm{Optimize}(\theta_G, \nabla_{\theta_G} \mathcal{L}_{\mathrm{GAN}} + \lambda_{\mathrm{MR}} \mathcal{L}_{\mathrm{MR}})$

---

---

**Algorithm 5** Generator update in Gaussian proxy MR$_1$-GAN

---

**Require:** Generator $G$, discriminator $D$, pre-trained predictor $P$, proxy MR loss coefficient $\lambda_{\mathrm{pMR}}$
**Require:** input $x$, ground truth $y$, the number of samples $K$
 1: **for** $i = 1$ **to** $K$ **do**
 2:    $z_i \leftarrow \mathrm{GenerateNoise}()$
 3:    $\tilde{y}_i \leftarrow G(x, z_i)$ {Generate $K$ samples}
 4: **end for**
 5: $\hat{\mu} \leftarrow P(x)$ {Predicted mean}
 6: $\tilde{\mu} \leftarrow \frac{1}{K} \sum_{i=1}^{K} \tilde{y}_i$ {Sample mean}
 7: $\mathcal{L}_{\mathrm{pMR}} \leftarrow (\hat{\mu} - \tilde{\mu})^2$
 8: $\mathcal{L}_{\mathrm{GAN}} \leftarrow \frac{1}{K} \sum_{i=1}^{K} -\log D(x, y_i)$
 9: $\theta_G \leftarrow \mathrm{Optimize}(\theta_G, \nabla_{\theta_G} \mathcal{L}_{\mathrm{GAN}} + \lambda_{\mathrm{pMR}} \mathcal{L}_{\mathrm{pMR}})$

---

---

**Algorithm 6** Generator update in Gaussian proxy MR$_2$-GAN

---

**Require:** Generator $G$, discriminator $D$, pre-trained predictor $P$, proxy MR loss coefficient $\lambda_{\mathrm{pMR}}$
**Require:** input $x$, ground truth $y$, the number of samples $K$
 1: **for** $i = 1$ **to** $K$ **do**
 2:    $z_i \leftarrow \mathrm{GenerateNoise}()$
 3:    $\tilde{y}_i \leftarrow G(x, z_i)$ {Generate $K$ samples}
 4: **end for**
 5: $\hat{\mu}, \hat{\sigma}^2 \leftarrow P(x)$ {Predicted mean and variance}
 6: $\tilde{\mu} \leftarrow \frac{1}{K} \sum_{i=1}^{K} \tilde{y}_i$ {Sample mean}
 7: $\tilde{\sigma}^2 \leftarrow \frac{1}{K-1} \sum_{i=1}^{K} (\tilde{y}_i - \tilde{\mu})^2$ {Sample variance}
 8: $\mathcal{L}_{\mathrm{pMR}} \leftarrow (\hat{\mu} - \tilde{\mu})^2 + (\hat{\sigma}^2 - \tilde{\sigma}^2)^2$
 9: $\mathcal{L}_{\mathrm{GAN}} \leftarrow \frac{1}{K} \sum_{i=1}^{K} -\log D(x, y_i)$
10: $\theta_G \leftarrow \mathrm{Optimize}(\theta_G, \nabla_{\theta_G} \mathcal{L}_{\mathrm{GAN}} + \lambda_{\mathrm{pMR}} \mathcal{L}_{\mathrm{pMR}})$

---

---

**Algorithm 7** Generator update in Laplace proxy $\text{MR}_1$-GAN

---

**Require:** Generator $G$, discriminator $D$, pre-trained predictor $P$, proxy MR loss coefficient $\lambda_{\text{pMR}}$
**Require:** input $x$, ground truth $y$, the number of samples $K$
1: **for** $i = 1$ **to** $K$ **do**
2:     $z_i \leftarrow \text{GenerateNoise}()$
3:     $\tilde{y}_i \leftarrow G(x, z_i)$ {Generate $K$ samples}
4: **end for**
5: $\hat{m} \leftarrow P(x)$ {Predicted median}
6: $\tilde{m} \leftarrow \text{med}(\tilde{y}_{1:K})$ {Sample median}
7: $t_{1:K} = \text{detach}(\tilde{y}_{1:K} + (\hat{m} - \tilde{m}))$
8: $\mathcal{L}_{\text{pMR}} \leftarrow \frac{1}{K} \sum_{i=1}^{K} (t_i - \tilde{y}_i)^2$
9: $\mathcal{L}_{\text{GAN}} \leftarrow \frac{1}{K} \sum_{i=1}^{K} -\log D(x, y_i)$
10: $\theta_G \leftarrow \text{Optimize}(\theta_G, \nabla_{\theta_G} \mathcal{L}_{\text{GAN}} + \lambda_{\text{pMR}} \mathcal{L}_{\text{pMR}})$

---

---

**Algorithm 8** Generator update in Laplace proxy $\text{MR}_2$-GAN

---

**Require:** Generator $G$, discriminator $D$, pre-trained predictor $P$, proxy MR loss coefficient $\lambda_{\text{pMR}}$
**Require:** input $x$, ground truth $y$, the number of samples $K$
1: **for** $i = 1$ **to** $K$ **do**
2:     $z_i \leftarrow \text{GenerateNoise}()$
3:     $\tilde{y}_i \leftarrow G(x, z_i)$ {Generate $K$ samples}
4: **end for**
5: $\hat{m}, \hat{b} \leftarrow P(x)$ {Predicted median and MAD}
6: $\tilde{m} \leftarrow \text{med}(\tilde{y}_{1:K})$ {Sample median}
7: $\tilde{b} \leftarrow \frac{1}{K} \sum_{i=1}^{K} |\tilde{y}_i - \tilde{m}|$ {Sample MAD}
8: $t_{1:K} = \text{detach}(\tilde{y}_{1:K} + (\hat{m} - \tilde{m}))$
9: $\mathcal{L}_{\text{pMR}} \leftarrow \frac{1}{K} \sum_{i=1}^{K} (t_i - \tilde{y}_i)^2 + (\hat{b} - \tilde{b})^2$
10: $\mathcal{L}_{\text{GAN}} \leftarrow \frac{1}{K} \sum_{i=1}^{K} -\log D(x, y_i)$
11: $\theta_G \leftarrow \text{Optimize}(\theta_G, \nabla_{\theta_G} \mathcal{L}_{\text{GAN}} + \lambda_{\text{pMR}} \mathcal{L}_{\text{pMR}})$

---

## B  TRAINING DETAILS

### B.1  COMMON CONFIGURATIONS

We use PyTorch for the implementation of our methods. In every experiment, we use AMSGrad optimizer (Reddi et al., 2018) with LR $= 10^{-4}, \beta_1 = 0.5, \beta_2 = 0.999$. We use the weight decay of a rate $10^{-4}$ and the gradient clipping by a value 0.5. In case of proxy MR-GAN, we train the predictor until it is overfitted, and use the checkpoint with the lowest validation loss. The weight of GAN loss is fixed to 1 in all cases.

**Convergence speed**. Our methods need more training steps (about $1.5\times$) to generate high-quality images compared to those with the reconstruction loss. This is an expectable behavior because our methods train the model to generate a much wider range of outputs.

**Training stability**. The proxy MR loss is similar to the reconstruction loss in terms of training stability. Our methods work stably with a large range of hyperparameter $\lambda$. For example, the coefficient of the proxy MR loss can be set across several orders of magnitude (from tens to thousands) with similar results. However, as noted, the MR loss is unstable compared to the proxy MR loss.

### B.2  PIX2PIX

Our Pix2Pix variant is based on the U-net generator from
`https://github.com/junyanz/pytorch-CycleGAN-and-pix2pix`.

- Noise input: We concatenate Gaussian noise tensors of size $H \times W \times 32$ at the $1 \times 1, 2 \times 2$, $4 \times 4$ feature map of the decoder. Each element in the noise tensors are independently sampled from $\mathcal{N}(0, 1)$.

- Input normalization: We normalize the inputs so that each channel has a zero-mean and a unit variance.

- Batch sizes: We use 16 for the discriminator and the predictor and 8 for the generator. When training the generator, we generate 10 samples for each input, therefore its total batch size is 80.

- Loss weights: We set $\lambda_{\mathrm{MR}} = \lambda_{\mathrm{pMR}} = 10$. For the baseline, we use $\ell_1$ loss as the reconstruction loss and set $\lambda_{\ell_1} = 100$.

- Update ratio: We update generator once per every discriminator update.

### B.3  SRGAN

Our SRGAN variant is based on the PyTorch implementation of SRGAN from
`https://github.com/zijundeng/SRGAN`.

- Noise input: We concatenate Gaussian noise tensor of size $H \times W \times 16$ at each input of the residual blocks of the generator, except for the first and last convolution layers. Each element in the noise tensors are independently sampled from $\mathcal{N}(0, 1)$.

- Input normalization: We make $16 \times 16 \times 3$ input images' pixel values lie between -1 and 1. We do not further normalize them with their mean and standard deviation.

- Batch sizes: We use 32 for the discriminator and the predictor and 8 for the generator. When training the generator, we generate 24 samples for each input, and thus its total batch size is 192.

- Loss weights: We set $\lambda_{\mathrm{MR}} = 20$ and $\lambda_{\mathrm{pMR}} = 2400$. For the baseline, we use $\ell_2$ loss as the reconstruction loss and set $\lambda_{\ell_2} = 1000$.

- Update ratio: We update generator five times per every discriminator update.

### B.4  GLCIC

We built our own PyTorch implementation of the GLCIC model.

- Noise input: We concatenate Gaussian noise tensor of size $H \times W \times 32$ at each input of the first and second dilated convolution layers. We also inject the noise to the convolution layer before the first dilated convolution layer. Each element in the noise tensors are independently sampled from $\mathcal{N}(0, 1)$.

- Input resizing and masking: We use square-cropped CelebA images and resize them to $128 \times 128$. For masking, we randomly generate a hole of size between 48 and 64 and fill it with the average pixel value of the entire training dataset.

- Input normalization: We make $128 \times 128 \times 3$ input images' pixel values lie between 0 and 1. We do not further normalize them with their mean and standard deviation.

- Batch sizes: We use 16 for the discriminator and the predictor and 8 for the generator. When training the generator, we generate 12 samples for each input, therefore its total batch size is 96.

- Loss weights: For GLCIC, we tested Gaussian proxy $MR_2$ loss and MR lossess. We successfully trained GLCIC with the Gaussian proxy $MR_2$ loss using $\lambda_{\mathrm{pMR}} = 1000$. However, we could not find any working setting for the MR loss. For the baseline model, we use $\ell_2$ loss for the reconstruction loss and set $\lambda_{\ell_2} = 100$.

- Update ratio: We update generator three times per every discriminator update.

## C   PREVENTIVE EFFECTS ON THE MODE COLLAPSE

Our MR and proxy MR losses have preventive effect on the mode collapse. Figure 4 shows toy experiments of unconditional generation on synthetic 2D data which is hard to learn with GANs due to the mode collapse. We train a simple 3-layer MLP with different objectives. When trained only with GAN loss, the model captures only one mode as shown in figure 4b. Adding $\ell_2$ loss cannot fix this issue either as in figure 4c. In contrast, all four of our methods (figure 4g-4f) prevent the mode collapse and successfully capture all eight modes. Notice that even the simpler variants $MR_1$ and proxy $MR_1$ losses effectively keep the model from the mode collapse. Intuitively, if the generated samples are biased toward a single mode, their statistics, *e.g.* mean or variance, deviate from real statistics. Our methods penalize such deviations, thereby reducing the mode collapse significantly. Although we restrict the scope of this paper to conditional generation tasks, these toy experiments show that our methods have a potential to mitigate the mode collapse and stabilize training even for unconditional generation tasks.

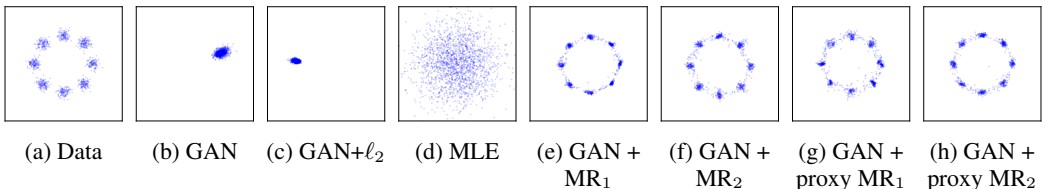

|     (a) Data     |    (b) GAN    |  (c) GAN+$\ell_2$  |  (d) MLE  |  (e) GAN + $MR_1$  |  (f) GAN + $MR_2$  |  (g) GAN + proxy $MR_1$  |  (h) GAN + proxy $MR_2$  |

Figure 4: Experiments on a synthetic 2D dataset. (a) The data distribution. (b)(c) Using the GAN loss alone or with $\ell_2$ loss results in the mode collapse. (d) Training a predictor with the MLE loss in Eq.(7) produces a Gaussian distribution with the mean and variance close to real distribution. The dots are samples from the Gaussian distribution parameterized by the outputs of the predictor. (e)-(h) Generators trained with our methods successfully capture all eight modes.

# D  GENERATED SAMPLES

## D.1  IMAGE-TO-IMAGE TRANSLATION (PIX2PIX)

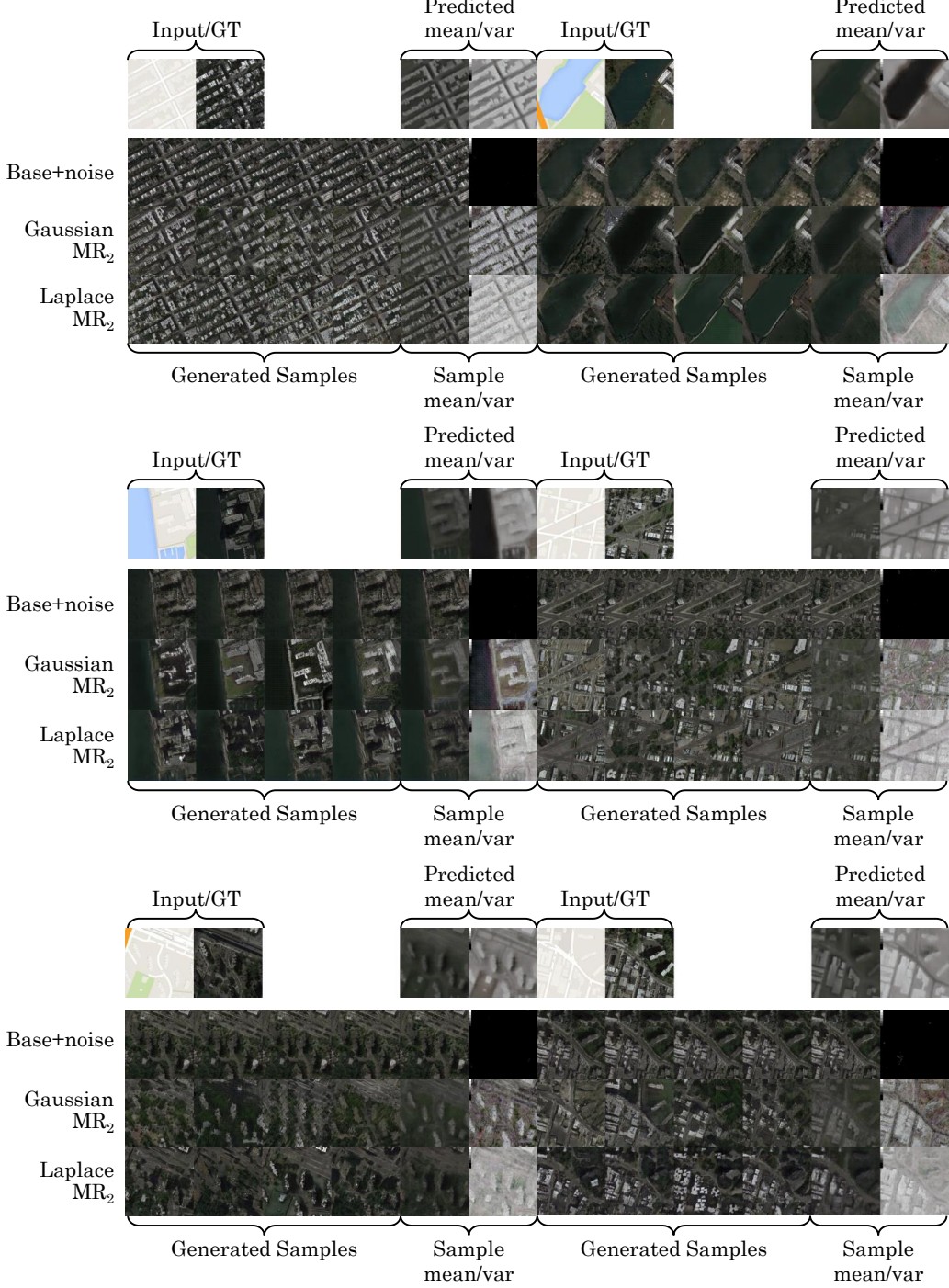

Figure 5: Comparison of our methods in Pix2Pix–Maps

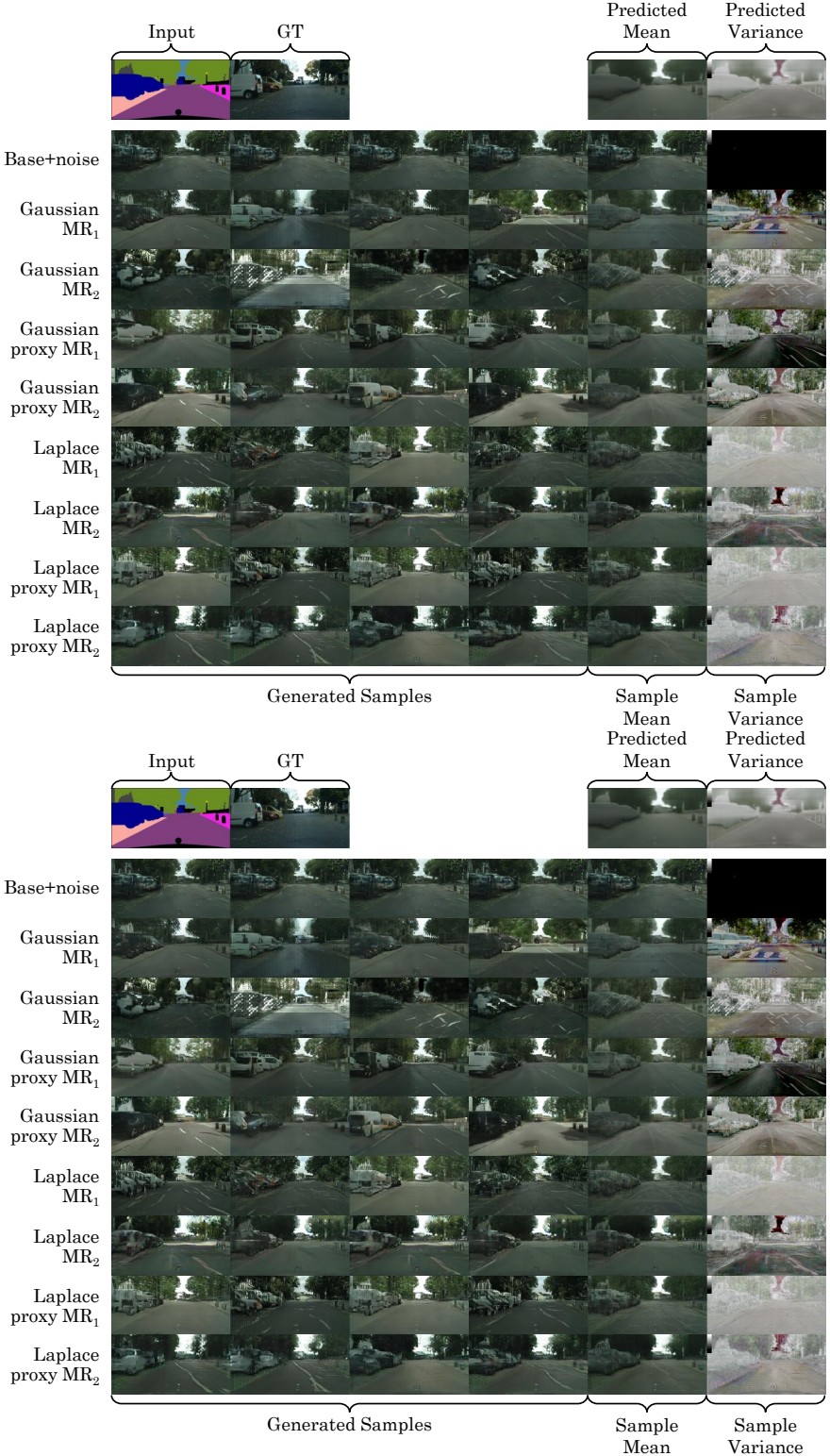

Figure 6: Comparison of our methods in Pix2Pix–Cityscapes

## D.2 Super-Resolution (SRGAN)

The first rows of following images are composed of input, ground-truth, predicted mean, sample mean, predicted variance, and sample variance. The other rows are generated samples.

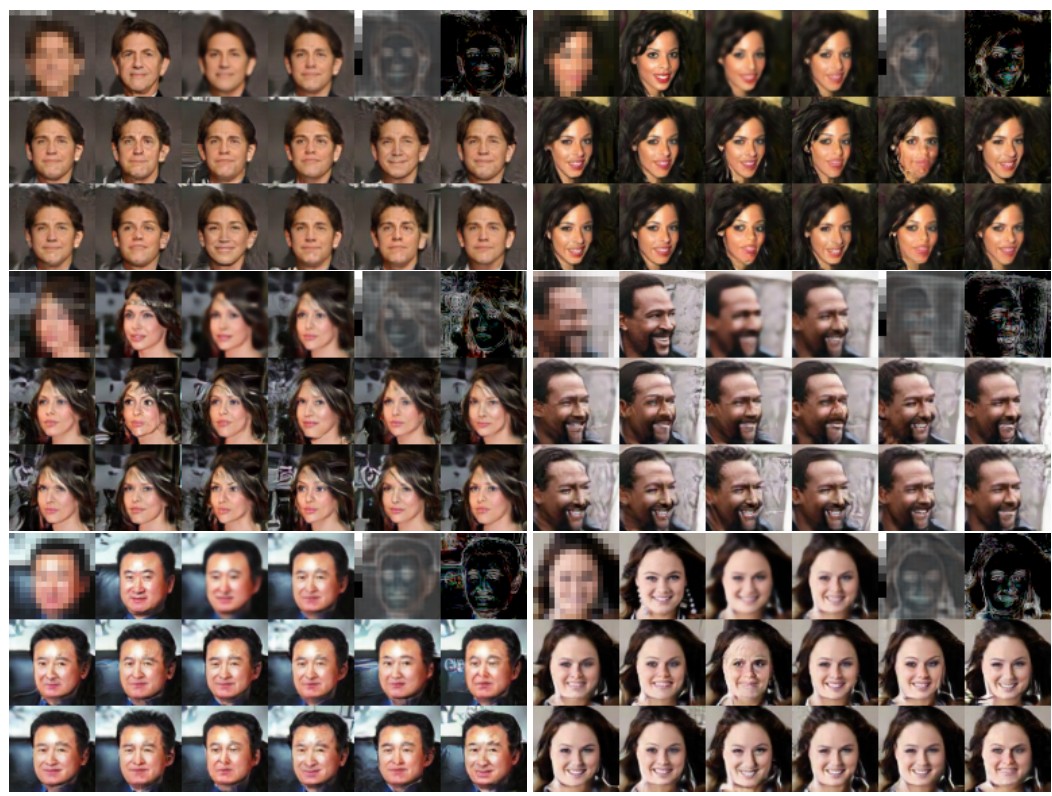

Figure 7: SRGAN–CelebA Gaussian proxy $MR_2$ loss (success cases).

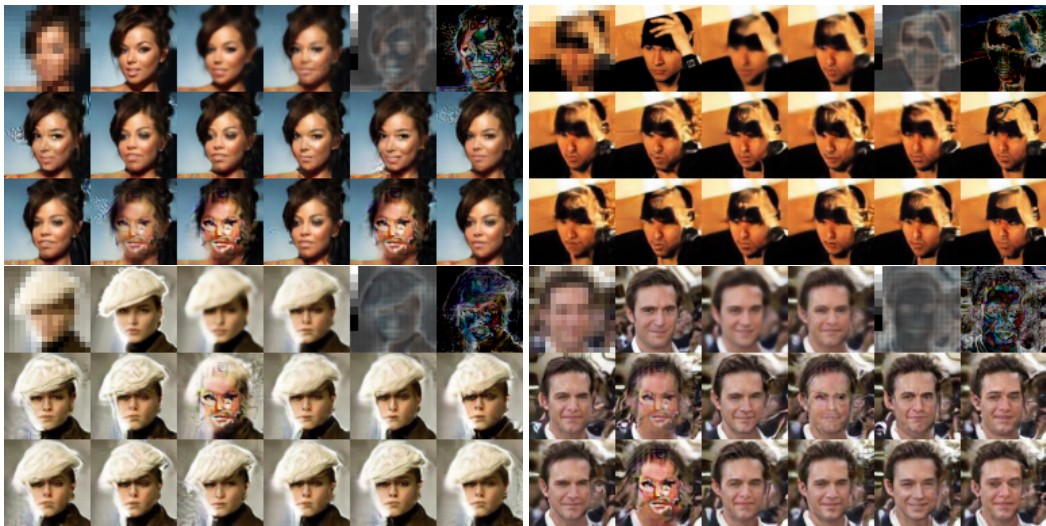

Figure 8: SRGAN–CelebA Gaussian proxy $MR_2$ loss (failure cases).

## D.3 IMAGE INPAINTING (GLCIC)

This section shows the results of GLCIC–CelebA task. The images are shown in the same manner as the previous section.

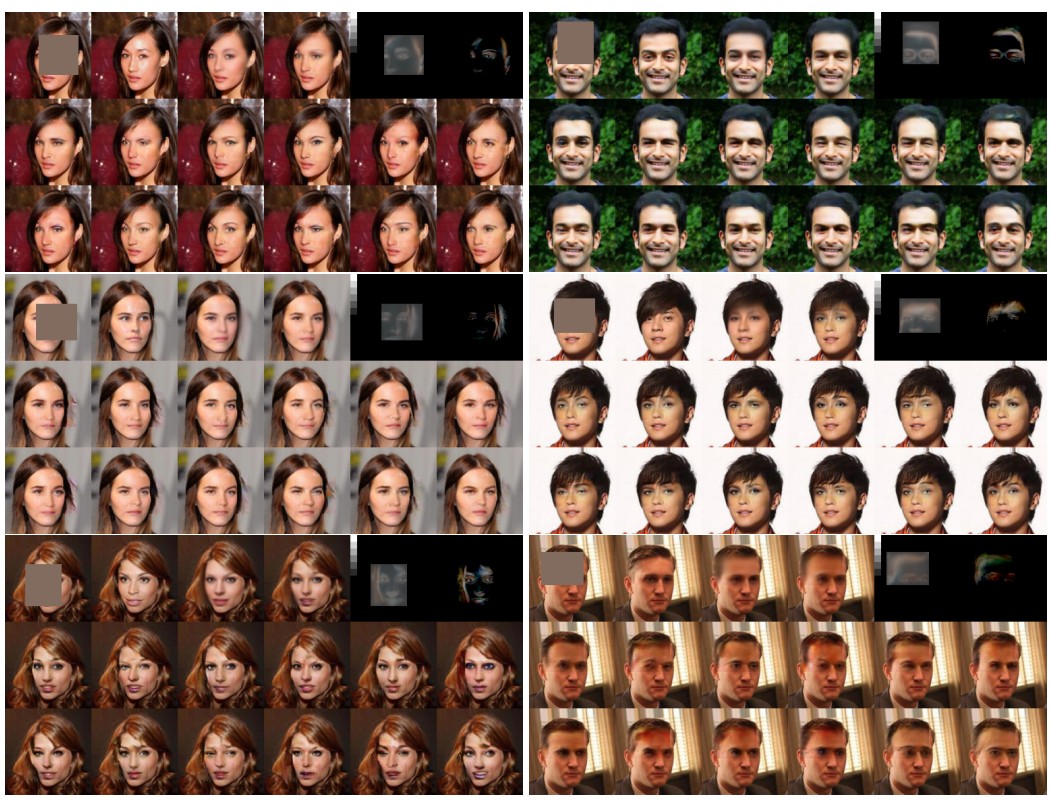

Figure 9: GLCIC–CelebA Gaussian proxy MR$_2$ loss (success cases).

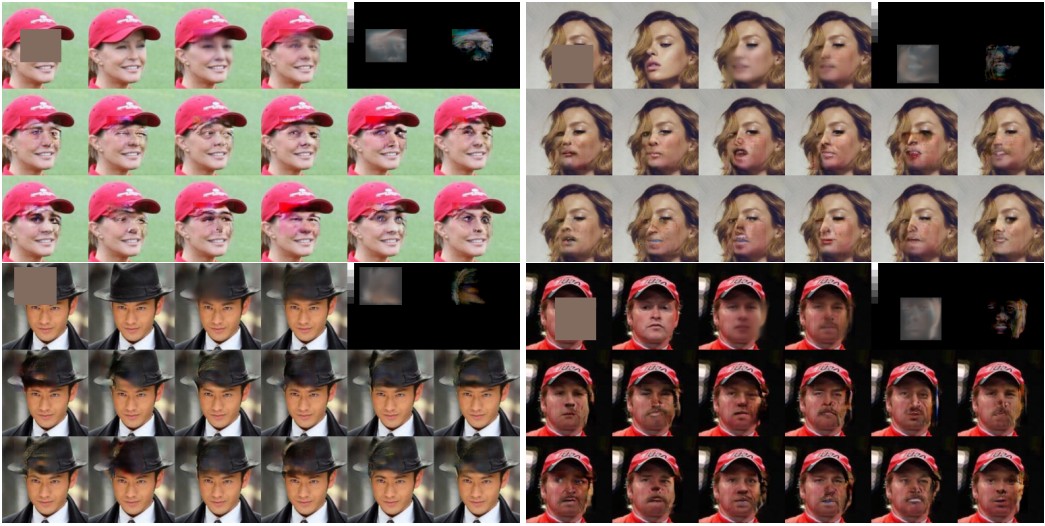

Figure 10: GLCIC–CelebA Gaussian proxy MR$_2$ loss (failure cases).

## E    MISMATCH BETWEEN $\ell_1$ LOSS AND GAN LOSS

In section 4.4, we showed that if there exists some $x$ such that $\mathrm{Var}(y|x) > 0$, there is no generator optimal for both $\ell_2$ loss and GAN loss. On the other hand, $\ell_1$ loss has some exceptional cases where the generator can minimize GAN loss and $\ell_1$ loss at the same time. We identify what the cases are and explain why such cases are rare.

To begin with, $\ell_1$ loss is decomposed as follows:

$$
\begin{aligned}
\mathcal{L}_1 &= \mathbb{E}_{x,y,z}\big[|y - G(x,z)|\big] \\
&= \mathbb{E}_{x,z}\Big[\mathbb{E}_y\big[|y - G(x,z)|\big]\Big] \\
&= \mathbb{E}_{x,z}\bigg[\int p(y|x)|y - G(x,z)|dy\bigg] \\
&= \mathbb{E}_{x,z}\bigg[\int_{-\infty}^{G(x,z)} p(y|x)(G(x,z) - y)dy + \int_{G(x,z)}^{\infty} p(y|x)(y - G(x,z))dy\bigg]
\end{aligned}
$$

To minimize $\ell_1$ loss, we need the gradient w.r.t. $G(x,z)$ to be zero for all $x$ and $z$ that $p(x) > 0$ and $p(z) > 0$. Note that this is a sufficient condition for minimum since the $\ell_1$ loss is convex.

$$
\frac{\partial \mathcal{L}_1}{\partial G(x,z)} = \int_{-\infty}^{G(x,z)} p(y|x)dy - \int_{G(x,z)}^{\infty} p(y|x)dy = 2\int_{-\infty}^{G(x,z)} p(y|x)dy - 1 = 0 \tag{18}
$$

$$
\int_{-\infty}^{G(x,z)} p(y|x)dy = \frac{1}{2} \tag{19}
$$

Therefore, $G(x,z)$ should be the conditional median to minimize the loss. Unlike $\ell_2$ loss, there can be an *interval* of $G(x,z)$ that satisfies Eq.(19). Specifically, any value between interval $[a,b]$ is a conditional median if $\int_{-\infty}^{a} p(y|x)dy = \int_{-\infty}^{b} p(y|x)dy = \frac{1}{2}$. If every real data belongs to the interval of conditional median, then the generator can be optimal for both GAN loss and $\ell_1$ loss.

For instance, assume that there are only two discrete values of $y$ possible for any given $x$, say $-1$ and $1$, with probability $0.5$ for each. Then the interval of median becomes $[-1, 1]$, thus any $G(x,z)$ in the interval $[-1, 1]$ minimizes the $\ell_1$ loss to $1$. If the generated distribution is identical to the real distribution, *i.e.* generating $-1$ and $1$ with the probability of $0.5$, the generator is optimal w.r.t. both GAN loss and $\ell_1$ loss.

However, we note that such cases hardly occur. In order for such cases to happen, for any $x$, every $y$ with $p(y|x) > 0$ should be the conditional median, which is unlikely to happen in natural data such as images. Therefore, the set of optimal generators for $\ell_1$ loss is highly likely to be disjoint with the optimal set for GAN loss.

## F    EXPERIMENTS ON MORE COMBINATIONS OF LOSS FUNCTIONS

We present some more experiments on different combinations of loss functions. The following results are obtained in the same manner as section D.2 and section D.3.

Figure 11 shows the results when we train the Pix2Pix model only with our loss term (without the GAN loss). The samples of $MR_1$ and proxy $MR_1$ losses are almost similar to those with the reconstruction loss only. Since there is no reason to generate diverse images, the variances of the samples are near zero while the samples are hardly distinguishable from the output of the predictor. On the other hand, $MR_2$ and proxy $MR_2$ losses do generate diverse samples, although the variation styles are different from one another. Intriguingly, $MR_2$ loss incurs high-frequency variation patterns as the sample variance of $MR_2$ loss is much closer than proxy $MR_2$ loss to the predicted variance. In the case of proxy $MR_2$ loss, where the attraction for diversity is relatively mild, the samples show low-frequency variation.

Another experiment that we carry out is about the joint use of all loss terms: GAN loss, our losses and reconstruction loss. Specifically, we use the following objective with varying $\lambda_{\mathrm{Rec}}$ from $0$ to

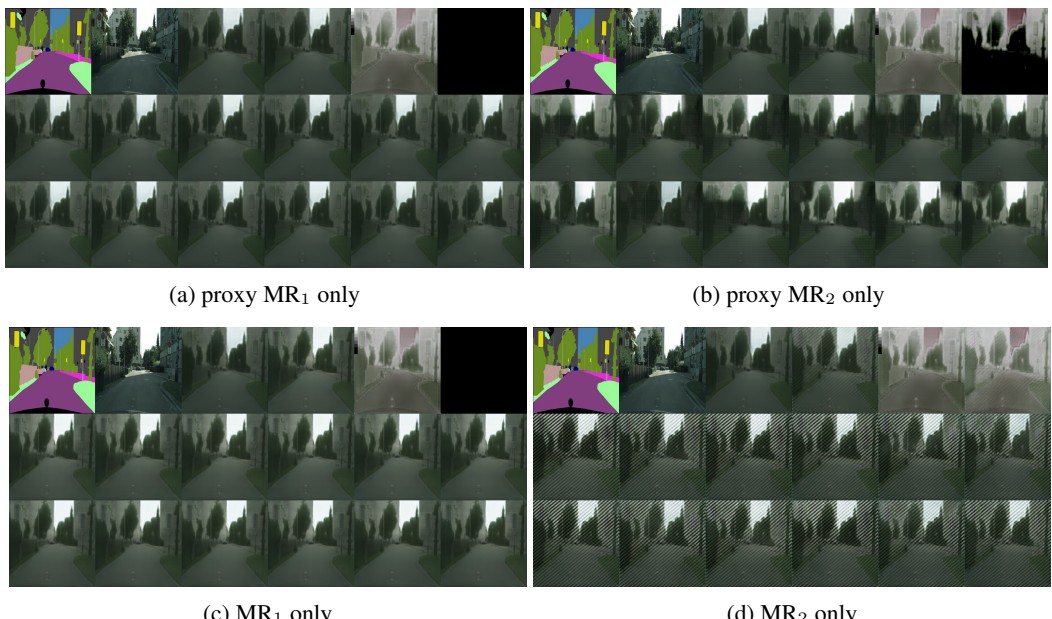

(a) proxy MR$_1$ only          (b) proxy MR$_2$ only

(c) MR$_1$ only            (d) MR$_2$ only

Figure 11: Ablation test using our loss terms only. The first row of each image is arranged in the order of input, ground truth, predicted mean, sample mean, predicted variance, and sample variance. The other two rows are generated samples. Without GAN loss, the results of proxy MR$_1$ loss and MR$_1$ loss are almost identical to the result of $\ell_2$ loss (predicted mean), while proxy MR$_2$ loss and MR$_2$ loss show variations with different styles.

100 to train the Pix2Pix model on Cityscapes dataset.

$$\mathcal{L} = \mathcal{L}_{\text{GAN}} + 10\mathcal{L}_{\text{pMR}} + \lambda_{\text{Rec}}\mathcal{L}_{\text{Rec}}$$

Figure 12 shows the results of the experiments. As $\lambda_{\text{Rec}}$ increases, the sample variance reduces. This confirms again that the reconstruction is the major cause of loss of variability. However, we find one interesting phenomenon regarding the quality of the samples. The sample quality deteriorates up to a certain value of $\lambda_{\text{Rec}}$, but gets back to normal as $\lambda_{\text{Rec}}$ further increases. It implies that either proxy MR loss or the reconstruction loss can find some high-quality local optima, but the joint use of them is not desirable.

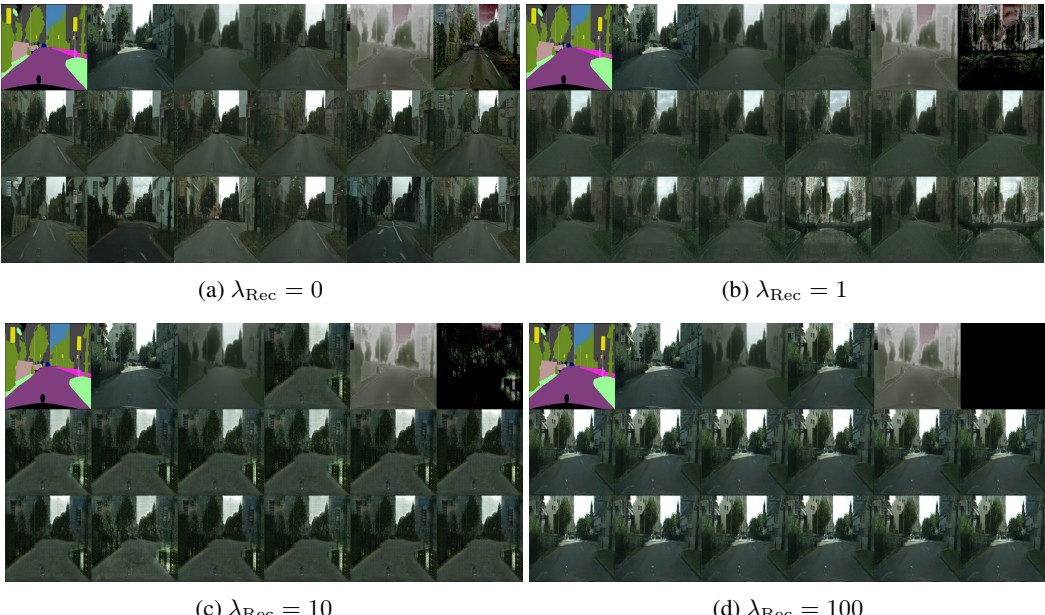

(a) $\lambda_{\mathrm{Rec}} = 0$        (b) $\lambda_{\mathrm{Rec}} = 1$

(c) $\lambda_{\mathrm{Rec}} = 10$        (d) $\lambda_{\mathrm{Rec}} = 100$

Figure 12: Joint use of GAN, proxy $\mathrm{MR}_2$, and reconstruction losses. Each image is presented in the same manner as figure 11. We fix the coefficients of GAN and proxy $\mathrm{MR}_2$ losses and test multiple values for the coefficient of reconstruction loss.

