# OpenReview forum: "Harmonizing Maximum Likelihood with GANs for Multimodal Conditional Generation"
_ICLR.cc/2019/Conference_

### Official Review · AnonReviewer3 · 2018-11-02
**Lack of novelty and weak theoretical results**

**Rating:** 4
**Confidence:** 5

**Review:**

The paper proposes a modification to the traditional conditional GAN objective (which minimizes GAN loss as well as either L1 or L2 pixel-wise reconstruction losses) in order to promote diverse, multimodal generation of images. The modification involves replacing the L1/L2 reconstruction loss -- which predicts the first moment of a pixel-wise gaussian/laplace (respectively) likelihood model assuming a constant spherical covariance matrix -- with a new objective that matches the first and second moments of a pixel-wise gaussian/laplace likelihood model with diagonal covariance matrix. Two models are proposed for matching the first and second moments - the first one involves using a separate network to predict the moments from data which are then used to match the generator’s empirical estimates of the moments (using K samples of generated images). The second involves directly matching the empirical moment estimates using monte carlo.

The paper makes use of a well-established idea - modeling pixel-wise image likelihood with a diagonal covariance matrix i.e. heteroscedastic variance (which, as explained in [1], is a way to learn data-dependent aleatoric uncertainty). Following [1], the usage of first and second moment prediction is also prevalent in recent deep generative models (for example, [2]) i.e. image likelihood models predict the per-pixel mean and variance in the L2 likelihood case, for optimizing Equation 4 from the paper. Recent work has also attempted to go beyond the assumption of a diagonal covariance matrix (for example, in [3] a band-diagonal covariance matrix is estimated). Hence, the only novel idea in the paper seems to be the method for matching the empirical estimates of the first and second moments over K samples. The motivation for doing this makes intuitive sense since diversity in generation is desired, which is also demonstrated in the results.

Section specific comments:
- The loss of modality of reconstruction loss (section 3.2) seems like something which doesn’t require the extent of mathematical and empirical detail presented in the paper. Several of the cited works already mention the pitfalls of using reconstruction loss.

- The analyses in section 4.4 are sound in derivation but not so much in the conclusions drawn. It is not clear that the lack of existence of a generator that is an optimal solution to the GAN and L2 loss (individually) implies that any learnt generator using GAN + L2 loss is suboptimal. More explanation on this part would help.

The paper is well written, presents a simple idea, complete with experiments for comparing diversity with competing methods. Some theoretical analyses do no directly support the proposition - e.g. sections 3.2 and 4.4 in my specific comments above. Hence, the claim that the proposed method prevents mode collapse (training stability) and gives diverse multi-modal predictions is supported by experiments and intuition for the method, but not so much theoretically. However, the major weakness of the paper is the lack of novelty of the core idea.

=== Update after rebuttal:
Having read through the other reviews and the author's rebuttal, I am unsatisfied with the rebuttal and I do not recommend accepting the paper. My rating has decreased accordingly.

The reasons for my recommendation, after discussion with other reviews, are -- (1) lack of novelty and (2) weak theoretical results (some justification of which was stated in my initial review above). Elaborating more on the second point, I would like to mention some points which came up during the discussion with other reviewers: The theoretical result which states that not using reconstruction loss given that multi-modal outputs are desired is a weaker result than proving that the proposed method is actually effective in what it is designed to do. There are empirical results to back that claim, but I strongly believe that the theoretical results fall short and feel out of place in the overall justification for the proposed method. This, along with my earlier point of lack of novelty are the basis for my decision.


References:
[1] Kendall, Alex, and Yarin Gal. "What uncertainties do we need in bayesian deep learning for computer vision?." Advances in neural information processing systems. 2017.
[2] Bloesch, M., Czarnowski, J., Clark, R., Leutenegger, S., & Davison, A. J. (2018). CodeSLAM-Learning a Compact, Optimisable Representation for Dense Visual SLAM. CVPR 2018.
[3] Dorta, G., Vicente, S., Agapito, L., Campbell, N. D., & Simpson, I. (2018, February). Structured Uncertainty Prediction Networks. In Proceedings of the IEEE Conference on Computer Vision and Pattern Recognition.

---

> ### Author Response · Authors · 2018-11-23
> **Answers to Reviewer 3**
>
> We are sincerely grateful for Reviewer 3’s thoughtful review. Please see blue fonts in the newly uploaded draft to check how our paper is updated.
>
> 1.  Novelty
> ===================================
> We believe that our work has significant novelties as follows:
>
> (1) To the best of our knowledge, our work is the first to formally criticize the use of reconstruction loss in conditional GANs. We also connect this problem to mode collapse (lose of multimodality). Of the prior works in conditional generation tasks, several papers empirically mention the loss of stochasticity in conditional GANs. However, they fail to analyze why this happens or propose what solutions can solve this problem. On the other hand, we reveal that the GAN loss and the reconstruction loss cannot coexist in harmony, and propose a solution to overcome this problem.
>
> (2) We propose alternatives to the reconstruction loss to greatly improve the multimodality of conditional GANs. As  Reviewer 3 pointed out, the components of our methods, MLE and moment matching, are well-established ideas. However, it is novel to combine them as a solution to the loss of multimodality in conditional generation. Furthermore, we think the simplicity of our methods is not a weakness but a strength, which makes our methods easily applicable to a wide range of conditional generation tasks.
>
> 2.  Specific comments on organization and drawn conclusions
> ===================================
> We reorganize section 3.2 and 4.4 to reflect Reviewer 3’s suggestion. Specifically, we simplify section 3.2 and move some content about reconstruction loss from 4.4 to 3.2.
>
> We agree with Reviewer 3 that the conclusion of section 4.4 may be rather over-stated. Our proof says that any generator cannot be optimal to both GAN and L2 loss simultaneously. It does not prove the generator is underperforming or suboptimal. Therefore, we remove the term ‘suboptimal’ and tone down the overall argument.
>
> We also cite the papers that Reviewer 3 suggested.

---

> ### Author Response · Authors · 2018-12-13
> **Re-clarification to Reviewer3’s Updated Review**
>
> We thank again the reviewer for the comments.
> However, we have the impression that some critics are unfair, imprecise and unhelpful; thus, hardly acceptable for us. Please see below why.
>
>
> 1. Novelty
> ===================================
> Reviewer3 raised again the concern about novelty in the updated review.
>
>
> In our rebuttal, we clarified that our work is the first to analyze why the use of reconstruction loss leads to the mode collapse (lose of multimodality) in conditional GANs. Our work is also the first to propose alternatives to the reconstruction loss which greatly improve the multimodality of conditional GANs without losing the visual fidelity of the output samples.
>
> Reviewer3 did not leave any comment on this clarification and failed to mention any specific works that undermine our novelty and how closely they are related to our work. In the initial review, reviewer3 referred several papers about variance prediction; however, these papers have no relation with conditional GANs or mode collapse.
>
> We sincerely ask reviewer3 to be specific and detailed on the claim that our work lacks novelty with proper ground.
>
>
> 2. Theoretical results
> ===================================
> “Proving that the proposed method is actually effective in what is designed to do”
> According to the modified review, reviewer3 seems to take the lack of proof that our model prevents mode collapse as a serious flaw in our work.
> However, we think reviewer3 largely misunderstood the key to our paper. Our methods have no multimodality-enhancing mechanism; instead, GANs are responsible for multimodality. Our methods are designed not to interrupt the GAN optimization and we proved it. The methods simply offer training stability without interference. Thus, the multimodality observed in our methods is inherent from GANs, and we pointed out that it is suppressed by the reconstruction loss in existing conditional GANs.
>
> Compared with a parallel submission to ICLR 2019 below, it becomes more obvious that we provide necessary proofs.
> Diversity-Sensitive Conditional Generative Adversarial Networks (https://openreview.net/forum?id=rJliMh09F7)
>
> Both papers share the same goal: multimodal generation in conditional GANs. However, the approaches are vastly different. Unlike our work, they add a regularization term to the loss while keeping the reconstruction loss. Their regularization term directly forces the model to generate diverse outputs. In this case, the proof that it facilitates diversity is necessary, so they present it. In contrast, we point out that the reconstruction loss conflicts with GANs in a way that reduces the output variance and proposes alternatives without such problem. Thus, we prove the problem of reconstruction loss and that our methods do not conflict with the GAN objective.
>
>
> 3. Suggestion for better reviewing process
> ===================================
> We carefully prepared for the rebuttal to answer to the initial review critics. However, we feel that our rebuttal is completely ignored because we cannot find in the updated review which specific question is not answered by our rebuttal and why our clarification cannot be the answer to the original review questions. We strongly believe the communications between authors and reviewers should be precise, specific and helpful to one another.

---

> > ### Comment · AnonReviewer3 · 2018-12-13
> > **Re: Re-clarification to Reviewer3’s Updated Review**
> >
> > Addressing the concerns that my updated review was imprecise or unhelpful and point (3) about the authors' rebuttal being ignored, I hope that the following points make it clear that the rebuttal was carefully considered in making my decision.
> >
> > 1. Regarding novelty
> >
> > My initial concern was regarding the novelty of the proposed method and not that of the criticism of the use of reconstruction loss in conditional GANs. The authors responded in the rebuttal that their paper has significant novelties in (1) formal criticism of the use of reconstruction loss and (2) their proposed method in the context of conditional generation tasks. Regarding the concern that I did not leave any comment on this response, my concern was regarding point (2), as the papers I cited, though not directly applied to conditional generation, demonstrate that the proposed method lacks novelty. However, the authors have stressed that I have ignored (1). This point was never a concern for me as I do agree that such a criticism of reconstruction loss has not been shown in prior work.
> >
> > My claim that the paper lacks novelty is specifically due to prior work such as CodeSLAM [1] using per-pixel mean and variance predictions. Regarding the point made in the authors' initial response that it is novel to combine these well-established ideas as a solution to loss of multimodality in conditional generation - it is indeed novel to combine or re-use existing ideas to new domains but in this case, the main approach proposed in the paper is solely a result of such re-use this significantly reduces the overall novelty of the paper.
> >
> >
> > 2. Regarding theoretical results
> >
> > “Our methods are designed not to interrupt the GAN optimization and we proved it”
> > Correct me if I’m wrong but I do not see any theoretical proof in the current revision of the paper showing that the proposed method does not interrupt GAN optimization. Section 3.2 in the current version seems to be the only theoretical proof which is criticizing the use reconstruction loss with GAN loss.
> >
> > “reviewer3 seems to take the lack of proof that our model prevents mode collapse as a serious flaw in our work”
> > For correctness, I was not looking for a specific proof about the proposed work preventing mode collapse. The question I would like to ask is - Given that the paper has shown that the optimizing reconstruction loss directly with GAN loss will lead to inevitable mode collapse, what is the proof that the proposed approach will not suffer the same fate? I agree that your proposed approach does not directly optimize the reconstruction loss, which gives intuition that the moment-matching approaches should not suffer from the same drawbacks. However, this does not constitute a theoretical result backing the claim that the proposed method will not interrupt GAN optimization.
> >
> > Comparing with Diversity-Sensitive Conditional Generative Adversarial Networks: I agree that their approach is vastly different, but the merits of that paper are completely different. Their proposed approach is novel and their analysis shows various perspectives of why their approach is effective. This again comes back to my point that this paper lacks theoretical analysis proving that the proposed method is effective.
> >
> > ”In contrast, we point out that the reconstruction loss conflicts with GANs in a way that reduces the output variance and proposes alternatives without such problem. Thus, we prove the problem of reconstruction loss and that our methods do not conflict with the GAN objective.”
> > My response to this point is the same as what I have stated earlier. I do not see how the second sentence follows from the first - the proof in Section 3.2 does not say anything about whether the proposed approach has any guarantees against mode collapse or conflicts with the GAN objective.
> >
> >
> > 3. Conclusion
> >
> > I wholeheartedly agree that the reviewers and authors must communicate in a precise and constructive way. I am happy to make my decision process transparent and continue this discussion.
> >
> >
> > [1] Bloesch, M., Czarnowski, J., Clark, R., Leutenegger, S., & Davison, A. J. (2018). CodeSLAM-Learning a Compact, Optimisable Representation for Dense Visual SLAM. CVPR 2018.

---

> > > ### Author Response · Authors · 2018-12-14
> > > **Re: Re: Re-clarification to Reviewer3’s Updated Review**
> > >
> > > We are deeply grateful to reviewer3 for a quick reply that reveals the detailed ground for the decision. Now we can understand the review much better to offer more focused answers to the concerns raised by reviewer3.
> > >
> > > 1. Novelty
> > > ===================================
> > > According to reviewer3’s clarification, per-pixel mean and variance prediction is the core of our methods, and thus our methods don’t have enough novelty compared to the cited papers.
> > >
> > > Although some of our methods involve mean and variance prediction, the key idea of our methods is matching the moments of the sample distribution to the maximum likelihood estimates of the real moments. As such, MLMM_1 and MCMLE_1, for example, do not use the variance prediction but achieve great diversity and quality.
> > >
> > > Note that our methods suggest two simple modifications to existing conditional GANs as final recipes; thus it would not be surprising that some previous work used similar techniques in other applications. However, we would like to emphasize that our methods are novel in the context of conditional GANs and mode collapse of GANs.
> > >
> > >
> > > 2. Theoretical results
> > > ===================================
> > > We would like to clarify that the proof that reviewer3 looks for is in section 4.4 not in section 3.2. During the rebuttal period, we reorganized section 3.2 and section 4.4 to reflect reviewer3’s comments and to streamline the logic. In the current draft, section 3.2 contains the proof about the conflict between the reconstruction loss and the GAN loss, while section 4.4 proves that our approach does not suffer from the same problem.

---

### Official Review · AnonReviewer2 · 2018-11-02
**An interesting paper in analyzing and improving model collapse problems in conditional GANs**

**Rating:** 7
**Confidence:** 3

**Review:**

This paper analyzes the model collapse problems on training conditional GANs and attribute it to the mismatch between GAN loss and reconstruction loss. This paper also proposes new types of reconstruction loss by measuring higher statistics for better multimodal conditional generation.

Pros:
1.	The analysis in Sec 4.4 is insightful, which partially explains the success of MLMM and MCMLE over previous method in generating diverse conditional outputs.
2.	The paper is well written and easy to follow.

Cons:
Analysis on the experiments is a little insufficient, as shown below.

I have some questions (and suggestions) about experiments.
1.	How does the training process affected by changing the reconstruction loss (e.g., how the training curve changes?)? Do MLMM and MCMLE converge slower or faster than the original ones? What about training stability?
2.	Why only MLMM_1 is not compared with other methods on SRGAN-celebA and GLCIC-A? From pix2pix cases it seems that Gaussian MLMM_1 performs much better than MLMM_{1/2}.

---

> ### Author Response · Authors · 2018-11-23
> **Answers to Reviewer 2**
>
> We thank Reviewer 2 for positive and constructive reviews. Please see blue fonts in the newly uploaded draft to check how our paper is updated.
>
> 1.  Convergence speed
> ===================================
> We observe that our methods need more training steps (about 1.5x) to generate high-quality images compared to that with the reconstruction loss. It might be obvious because our methods train the model to generate a much wider range of outputs. We add some comments to Appendix B.1 regarding the convergence speed.
>
> 2.  Training stability
> ===================================
> MLMM is similar to the reconstruction loss in terms of training stability. Encouragingly, our methods stably work with a large range of hyperparameter \lambda. For example, the loss coefficient of MLMM is settable across several orders of magnitude (from tens to thousands) with similar results. However, as noted in the paper, MCMLE is unstable compared to MLMM.
>
> 3.  Why only MLMM_1 is not compared
> ===================================
> Due to many combinations between our methods and tasks, we had to choose only a few of our methods for human evaluation. Although MLMM_1 and MLMM_{1/2} attained similar performance for all three tasks, we chose MLMM_{1/2} as the ‘default’ method because it better implements our idea - matching more statistics (i.e. not only means but also variances).

---

### Official Review · AnonReviewer1 · 2018-11-03
**Accept**

**Rating:** 8
**Confidence:** 4

**Review:**

The paper describes an alternative to L1/L2 errors (wrt output and one ground-truth example) that are used to augment adversarial losses when training conditional GANs. While these augmented losses are often needed to stabilize and guide GAN training, the authors argue that they also bias the optimization of the generator towards mode collapse. To address this, the method proposes two kinds of alternate losses--both of which essentially generate multiple sample outputs from the same input, fit these with a Gaussian distribution by computing the generating sample mean and variance, and try to maximize the likelihood of the true training output under this distribution. The paper provides theoretical and empirical analysis to show that the proposed approach leads to generators that produce samples that are both diverse and high-quality.

I think this is a good paper and solves an important problem---where one usually had to sacrifice diversity to obtain stable training by adding a reconstruction loss. I recommend acceptance.

An interesting ablation experiment might be to see what happens when one no longer includes the GAN loss and trains only with the MLMM or MCMLE losses, and compare this to training with only the L1/L2 losses. The other thing I'd like the authors to comment on are the potential shortcomings of using a simple un-correlated Gaussian to model the sample distributions. It seems that such a distribution may not capture the fact that multiple dimensions of the output (i.e., multiple pixel intensities) are not independent conditioned on the input. Perhaps, it may be worth exploring whether Gaussians with general co-variance matrices, or independent in some de-correlated space (learned from say simply the set of outputs) may increase the efficacy of these losses.

====Post-rebuttal

I've read the other reviews and retain my positive impression of the paper. I also appreciate that the authors have conducted additional experiments based on my (non-binding) suggestions---and the results are indeed interesting. I am upgrading my score accordingly.

---

> ### Author Response · Authors · 2018-11-23
> **Answers to Reviewer 1**
>
> We thank Reviewer 1 for your encouraging and constructive comments. Please see blue fonts in the newly uploaded draft to check how our paper is updated.
>
> 1.  Ablation experiments
> ===================================
> We carry out the ablation experiments and present the results in appendix G (page 22). The results are indeed interesting. When trained with MLMM_1 or MCMLE_1 only, the outputs are indistinguishable from those with the reconstruction loss only, since there is no variation-inducing term to generate diverse output. In the case of MLMM_{1/2} and MCMLE_{1/2}, the model shows high variation in the output. However, the patterns of the variations differ greatly. Specifically, MLMM_{1/2} shows variations in low-frequency while MCMLE_{1/2} shows those in high-frequency.
>
> We also add experiments of using GAN loss, MLMM_{1/2} loss, and reconstruction loss altogether. Whiling fixing the coefficient of GAN loss and MLMM loss to 1 and 10 respectively, we gradually increase the coefficient of reconstruction loss from 0 to 100. We find that the output variation decreases as the reconstruction loss increases. Interestingly, the sample quality is high when the reconstruction loss is absolutely zero or dominated by the MLMM loss. In contrast, the samples show poor quality when the reconstruction coefficient is 1 or 10. It seems that either method can assist the GAN loss to find visually appealing local optima but the joint use of them leads to a troublesome behavior.
>
> 2.  Shortcomings of un-correlated Gaussian
> ===================================
> This is a very interesting and profound question that may need to be further investigated in the future work. In summary, we believe that incorporating more statistics is not guaranteed to improve the performance, and un-correlated Gaussian may not be a bad choice.
>
> An ideal GAN loss can match with any kind of statistics since it minimizes the JS divergence between sample distribution and real distribution. In this sense, additional loss term should be regarded as a ‘guidance’ signal, while the key player is still the GAN loss. However, it is unclear whether a tighter guidance necessarily yields better outputs.
>
> Regarding the tightness of guidance, the loss terms can be ordered as follows:
> MLMM_1 = MCMLE_1 < MLMM_{1/2} = MCMLE_{1/2} < general covariance Gaussian.
>
> Interestingly, our qualitative evaluations show that MLMM_1 and MCMLE_1 generate comparable or even better outputs compared to MLMM_{1/2} and MCMLE_{1/2}. That is, matching means could be enough to guide GAN training in many cases. Adding more statistics may be helpful in some cases, but generally may not improve the performance. Moreover, we should consider the errors arising from the statistics prediction because a wrong estimation of statistics can even misguide the GAN training.
>
> Please see blue fonts in section 5.2 of the newly uploaded draft to check how our paper is updated.

---

### Public Comment · (anonymous) · 2018-11-10
**About suboptimal generator**

Hi,
I think this is an interesting work for improving the diversity of cGAN.
But I have some questions:
1. The analysis in section 4.4 give a proof to  mode collapse of some cGANs, such as pix2pix or UNIT. But the proof is not supported to the model that encode the laten representation to help generate images (Var(y|x,c)=0 is ok in this case), such as BicycleGAN or MUNIT. Right?

2. The diversity scores in Table 1.(a) are remarkable. It will be interesting if you can present more comparisons with BicycleGAN in different tasks.

---

> ### Author Response · Authors · 2018-11-23
> **Thank you for your interest in our paper.**
>
>
> 1. The scope of our proof
> ===================================
> That’s a great point. We have to make it clear in the draft. Our proof is confined to conditional GAN models with no explicit latent variable. Since the explicit latent variables provide the model with a vehicle that can represent variability and multimodality, our argument in section 4.4 may not be applicable to the models that explicitly encode latent variables. We add this discussion to the end of section 4.4.
>
> 2. BicycleGAN
> ===================================
> BicycleGAN has been applied to image-to-image translation, but not to image inpainting and super-resolution. Thus, we cannot find any standard implementation (or learned parameters) of BicycleGAN for the two tasks, which was the main reason why we did not report its results on the two tasks - image inpainting and super-resolution.

---

### Meta-Review · Area_Chair1 · 2018-12-08
**Intersting new loss function for cGANs**

**Confidence:** 3
**Recommendation:** Accept (Poster)

**Metareview:**

The paper presents new loss functions (which replace the reconstruction part) for the training of conditional GANs. Theoretical considerations and an empirical analysis show that the proposed loss can better handle multimodality of the target distribution than reconstruction based losses while being competitive in terms of image quality.